# Multiomic analysis of the Arabian camel (*Camelus dromedarius*) kidney reveals a role for cholesterol in water conservation

Fernando Alvira-Iraizoz [1,7✉], Benjamin T. Gillard[1,7], Panjiao Lin[1], Alex Paterson[1], Audrys G. Pauža[1], Mahmoud A. Ali[2], Ammar H. Alabsi [3], Pamela A. Burger [4], Naserddine Hamadi[5], Abdu Adem [2,6,8✉], David Murphy [1,8] & Michael P. Greenwood [1,8]

The Arabian camel (*Camelus dromedarius*) is the most important livestock animal in arid and semi-arid regions and provides basic necessities to millions of people. In the current context of climate change, there is renewed interest in the mechanisms that enable camelids to survive in arid conditions. Recent investigations described genomic signatures revealing evolutionary adaptations to desert environments. We now present a comprehensive catalogue of the transcriptomes and proteomes of the dromedary kidney and describe how gene expression is modulated as a consequence of chronic dehydration and acute rehydration. Our analyses suggested an enrichment of the cholesterol biosynthetic process and an over-representation of categories related to ion transport. Thus, we further validated differentially expressed genes with known roles in water conservation which are affected by changes in cholesterol levels. Our datasets suggest that suppression of cholesterol biosynthesis may facilitate water retention in the kidney by indirectly facilitating the AQP2-mediated water reabsorption.

[1] Molecular Neuroendocrinology Research Group, Bristol Medical School: Translational Health Sciences, University of Bristol, Bristol, UK. [2] Department of Pharmacology and Therapeutics, College of Medicine and Health Sciences, United Arab Emirates University, AL Ain, United Arab Emirates. [3] College of Medicine, Alfaisal University, Riyadh, Saudi Arabia. [4] Department of Interdisciplinary Life Sciences, Research Institute of Wildlife Ecology, Vetmeduni Vienna, Vienna, Austria. [5] Department of Life and Environmental Sciences, College of Natural and Health Sciences, Zayed University, Abu Dhabi, United Arab Emirates. [6] Department of Pharmacology and Therapeutics, College of Medicine and Health Sciences, Khalifa University, Abu Dhabi, United Arab Emirates. [7] These authors contributed equally: Fernando Alvira-Iraizoz, Benjamin T. Gillard. [8] These authors jointly supervised this work: Abdu Adem, David Murphy, Michael P. Greenwood. ✉email: f.alvirairaizoz@gmail.com; abdu.adem@ku.ac.ae

The one-humped Arabian camel (*Camelus dromedarius*) is the most important livestock animal in the arid and semi-arid regions of North and East Africa, the Arabian Peninsula and Iran, and continues to provide basic needs to millions of people[1,2]. Thought to have been domesticated 3000–6000 years ago in the Arabian Peninsula[1,3–6], the camel has been used as a beast of burden, for riding and sport, and to produce milk, meat and shelter, and they are still used today for the same purposes[3,7–10]. Indicative of its reliability in harsh environments is the existence of several breeds that have been favoured in different regions according to their intended roles, for instance; for milk, meat or wool production, or as racing animals[6,11–16].

Amongst Camelidae, llamas (genus Lama) and alpacas (genus Vicugna) are adapted to the high-altitude steppes of South America while dromedaries and Bactrian camels (*Camelus bactrianus*) occupy desert environments of the Old World[6,17]. Extensive evidence shows the impressive set of adaptations that allows the dromedary to thrive in such environments[1,3,18–20], despite sometimes needing to survive for weeks without access to water. Altogether, these behavioural and physiological adaptations ensure that water is never wasted. Dromedaries solely eat the leaves of plants[21], avoid exposure to direct sunlight when possible[8,22], restrict reproduction to the cooler winter season[3,23], and drink very large amounts of water when available to compensate for pre-existing fluid deficiency[3,8,24–27]. Further, drinking water is slowly absorbed to avoid osmotic shock[28]. A relatively low haematocrit, the shape of the erythrocytes[3] and a modified haemoglobin are also advantageous in withstanding dehydration[1]. An intricate nasal passage allows the dromedary to recover water during expiration, thus limiting water loss[29,30]. More importantly, water evaporates from the nasal surfaces to moisturise dry air at inspiration, thus, heat dissipates from the venous blood by evaporative heat loss and it cools as it passes along the evaporative surfaces of the nasal passage. It is then forced through the cavernous sinus. The thin walls that separate arterial from venous blood in the sinus allow for an efficient heat exchange, which decreases the temperature of the arterial blood supplying the brain[19]. This mechanism results in a brain temperature considerably lower than that of the body core. From a metabolic point of view, dromedaries exhibit an overwhelming capacity to build up fat reserves during favourable periods and mobilize them when food is scarce[21,31]. In addition, body temperature may fluctuate (heterothermia)[8,32–34] up to 7 °C between day and night in dehydrated animals[26] which saves about 5 litres of water per day that would otherwise be lost in sweat[1]. The excess heat is dissipated during the cooler night.

In terms of the animal's ability to conserve water, it is the kidney that is a key player through the production of highly concentrated urine[20,35,36]. As explained by Fenton and Knepper[37] in a mouse model, "the concentrating process can, over-simplifying, be divided into two elements": (1) an active transport of solutes from the Loop of Henle into the medullary interstitium to create an osmotic gradient and (2) water reabsorption from the collecting ducts and the thin descending limbs (tDL) of the Loop of Henle mediated by the brain hormone arginine vasopressin (AVP). In the outer medulla, the watertight thick ascending limb (TAL) of the Loop of Henle actively pumps NaCl into the interstitium creating an osmotic gradient. A similar gradient is found in the inner medulla, although urea excreted from the collecting ducts, and not NaCl, is thought to be responsible for the osmotic gradient in this segment[38]. In fact, it has been shown in rats producing maximally concentrated urine that the highest osmotic gradient is generated in the inner medulla which has an osmolality around nine times that of the cortex[39]. Subsequently, the water-permeable tDL and the collecting duct allow water to flow through the membrane to restore osmotic equilibrium. This process is repeated as the filtrate passes through the tubular lumen creating the countercurrent multiplication effect, which results in highly concentrated urine and effective water conservation[37,40–42].

In order to produce highly concentrated urine, the kidney must possess certain anatomical features, which vary according to the aridity of the habitat[43]. The relative medullary thickness and the medulla/cortex ratio of the one-humped Arabian camel, both indices of the length of the Loop of Henle and the vasa recta[43] and predictors of maximum concentrating ability[44,45], have been reported to be 7.89[46] and ~4/1[1,20,36]. In humans, these values are 3.0[47] and ~2.7/1[48], respectively. This large medullary interface is lined by simple, low cuboidal epithelium which enhances the recycling of urea and water. Moreover, this inner medullary arrangement consists of vasa recta, tDL and TAL of Loop of Henle and collecting ducts, all essential for water conservation and urine concentration[43,49,50]. It is in the medulla where the most important steps towards producing highly concentrated urine take place.

In the current context of advancing desertification and climate change, there is renewed interest in the physiology of camels, particularly, the adaptive mechanisms that enable their survival in arid conditions. Moreover, state-of-the-art molecular biology techniques allow to investigate the genetic basis of these remarkable physiological adaptations. Here we present a comprehensive catalogue of the transcriptomes and proteomes of the dromedary kidney cortex and medulla. We further describe how gene expression profiles are modulated as a consequence of chronic dehydration and subsequent acute rehydration. Our data are strongly suggestive of a hypothesis that implicates the regulation of the level of cholesterol as playing a key role in the ability of the dromedary kidney to produce highly concentrated urine during water deprivation.

## Results

**Analysis of differential gene expression.** After an experimental period of 23 days (Fig. 1a) we analysed the transcriptomes and proteomes of the one-humped Arabian camel kidney cortex and medulla (Supplementary Data 1). Illumina Sequencing produced ~30 million reads per sample. We used DESeq2 for differential expression analysis with a significance threshold of 0.05 (Benjamini–Hochberg adjusted). According to our RNAseq data, the expression profiles of the cortex and the medulla were well-differentiated with respect to each other (Fig. 1b). However, grouping by condition was not as clear. Medulla samples clustered closely together. We got similar results for the cortex, except for samples from dehydrated animals which were well-differentiated (Fig. 1b). Despite the full set of expressed genes not clustering the samples by condition, we were able to catalogue a large number of genes with significant ($p_{adj} < 0.05$) differential expression between control, dehydrated and rehydrated animals. Moreover, heatmaps of differentially expressed genes (DEGs) showed that this subset of genes tightly clustered by condition in both cortex and medulla (Fig. 1c, d).

The DEGs identified in the cortex and the medulla were very different with respect to each other; 642 DEGs were identified as being expressed only in the cortex and 505 DEGs were found only in the medulla. Only 135 DEGs were co-regulated in both tissues in at least one condition (Fig. 1e, f and Supplementary Data 2).

Analyses of tissue-specific differential expression across conditions showed that, in the cortex, 701 DEGs emerged during dehydration compared to controls. Six hundred and fifty-six of these DEGs returned to control expression levels following rehydration while 45 DEGs remained differentially co-regulated in both conditions compared to controls. An additional 76 DEGs emerged during rehydration compared to controls (Fig. 1f, g).

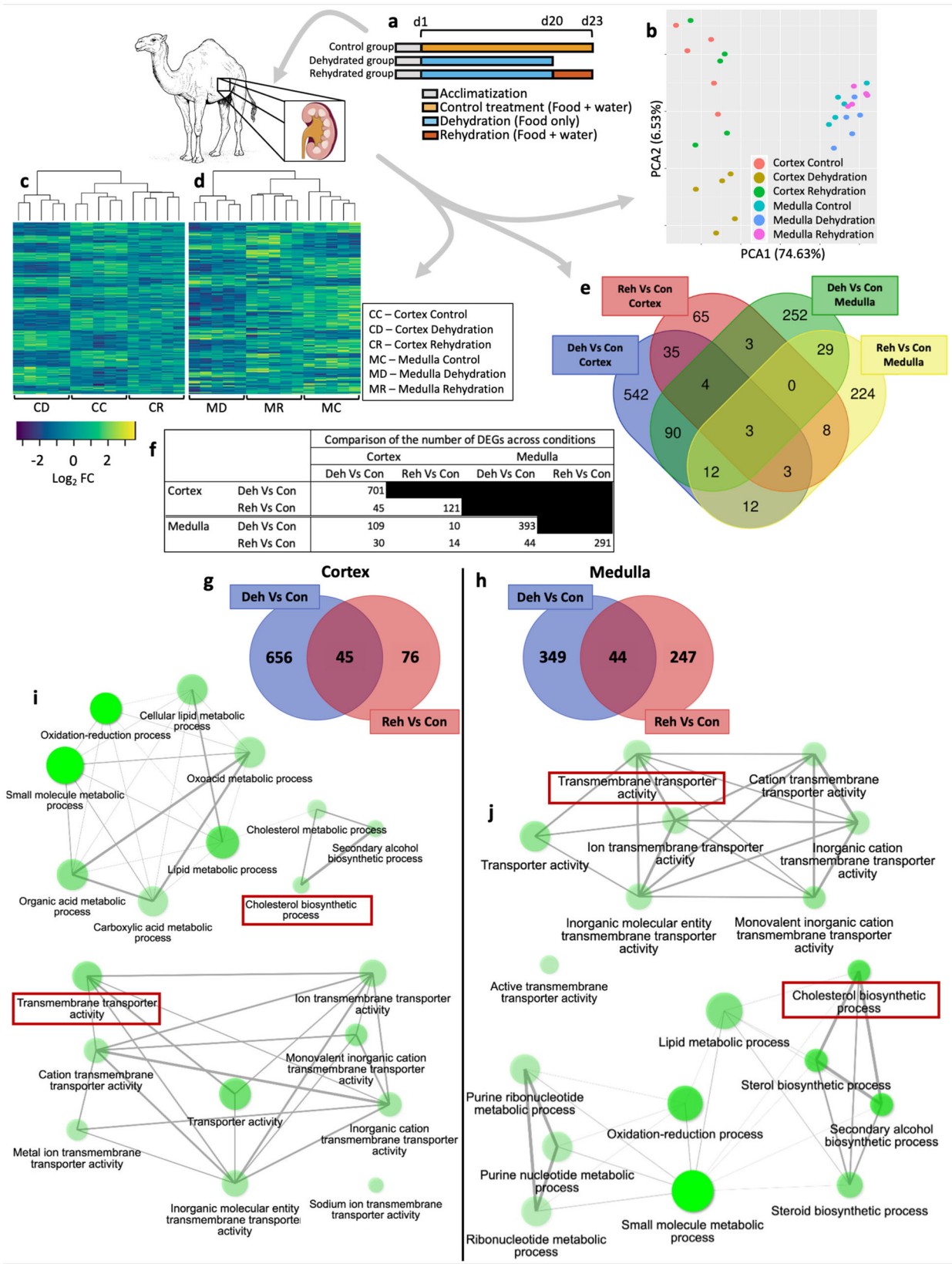

CC – Cortex Control
CD – Cortex Dehydration
CR – Cortex Rehydration
MC – Medulla Control
MD – Medulla Dehydration
MR – Medulla Rehydration

| | | Comparison of the number of DEGs across conditions | | | |
|---|---|---|---|---|---|
| | | Cortex | | Medulla | |
| | | Deh Vs Con | Reh Vs Con | Deh Vs Con | Reh Vs Con |
| Cortex | Deh Vs Con | 701 | | | |
| | Reh Vs Con | 45 | 121 | | |
| Medulla | Deh Vs Con | 109 | 10 | 393 | |
| | Reh Vs Con | 30 | 14 | 44 | 291 |

Further, 429 DEGs showed significant changes in rehydration when compared to dehydrated samples (Supplementary Data 2). In the medulla, 393 DEGs emerged during dehydration compared to controls. Three hundred and forty-nine of these DEGs returned to control expression levels during rehydration while 44 DEGs remained differentially co-regulated in both conditions.

Another 247 different DEGs emerged following rehydration compared to controls (Fig. 1f, h and Supplementary Data 2). In the medulla, 592 DEGs showed significant changes between rehydrated and dehydrated samples (Supplementary Data 2).

Our data showed that 1139 genes were differentially expressed in the dromedary kidney during dehydration compared to

**Fig. 1 Experimental design and analysis of the transcriptomes of the one-humped Arabian camel kidney during severe dehydration and acute rehydration. a** Experimental protocol for each experimental group. **b** PCA plot displaying the clustering of the samples based on their expression profiles where all expressed genes are included. **c**, **d** Heatmaps including all differentially expressed genes (DEGs) identified in the cortex (**c**) and medulla (**d**) across conditions show clear clustering of differentially expressed genes by condition. **e** Venn plot displaying the overlap of differentially expressed genes identified in dehydration and rehydration compared to control in cortex and medulla. **f** Absolute number of DEGs displayed in panels (**e**), (**g**) and (**h**). **g**, **h** Venn plots showing the distribution of DEGs during dehydration and rehydration compared to controls in the kidney cortex (**g**) and medulla (**h**) of the camel. Number of genes differentially expressed solely in dehydration compared to control (blue), in rehydration compared to control (light red) and those differentially co-regulated in both conditions (dark red) are shown. (**i**), **j** Gene Ontology (GO) networks displaying a subset of the most significantly enriched ($p < 0.05$) biological processes and molecular functions in the kidney cortex (**i**) and medulla (**j**) during dehydration. Two nodes are connected if they share 20% or more genes and thicker edges represent more overlapped genes, darker nodes are more significantly enriched gene sets and bigger nodes represent larger gene sets.

controls. Of all DEGs identified in dehydrated animals, 88% and 30%, returned to control expression levels in the cortex and the medulla, respectively. We identified a larger number of DEGs during dehydration than during rehydration in both tissues.

**GO analyses of DEGs revealed an overrepresentation of lipid/ cholesterol metabolism and ion transmembrane transport activity in the kidney.** We used ShinyGO[51] for Gene Set Enrichment Analysis to identify biological processes and/or molecular functions enriched during dehydration and rehydration in the camel kidney. We identified several significantly enriched ($p < 0.05$) Gene Ontology (GO) terms in the cortex and medulla (Supplementary Data 3), respectively. Interestingly, GO terms associated with "lipid metabolic processes" (GO:0006629), the "cholesterol biosynthetic process" (GO:0006695) and "trans-membrane transporter activity" (GO:0022857) were over-represented in both tissues during dehydration (Fig. 1i, j).

**Expression profiles.** We identified six different expression profiles amongst all DEGs by plotting $\log_2$FC calculated for every DEG and condition as a time series (Fig. 2 and Supplementary Data 4). The majority of the DEGs (71%) followed expression profiles A and B (Fig. 2a, b). Expression profiles C (Fig. 2c) and D (Fig. 2d) represented 14% and 9% of DEGs, respectively. Finally, only 2% and 4% of DEGs showed type-E and type-F profiles, respectively (Fig. 2e, f). In general, clustering by condition was better defined in the cortex than in the medulla. However, samples from dehydrated camels are well-differentiated and clustered together in most cases (Fig. 2). Genes coding for enzymes involved in the cholesterol biosynthetic process and important ion transporters, strongly overrepresented in the GO analyses as mentioned above, displayed type-B and type-A expression profiles, respectively. DEGs following these expression profiles showed good clustering by condition. Thus, we further confirmed that these changes in gene expression were due to changes in the hydration status of the camels. Other expression profiles are possible in a control-dehydration-rehydration 3-point time series comparison, but these were not represented in our RNAseq data except for 3 isolated DEGs in the medulla which were significantly downregulated during dehydration and significantly upregulated during rehydration compared to controls. No relevant functions associated with dehydration or rehydration could be attributed to these genes.

**Analysis of differential protein abundance.** Differentially expressed proteins (DEPs) distinctly clustered by condition in the cortex (Fig. 3a). In the medulla, differentiation between DEPs from control and dehydrated samples was not as clear (Fig. 3b). Differential abundance analysis showed that 826 DEPs were described only in the cortex and 1368 only in the medulla. Another 798 DEPs were co-regulated in both tissues in at least

one condition (Fig. 3c, d and Supplementary Data 5). Furthermore, dehydration induced the differential expression of 1292 proteins in the cortex compared to controls. Four hundred and forty of these DEPs returned to control expression levels during rehydration while 332 different DEPs emerged in rehydration compared to controls. A total of 852 DEPs remained differentially co-regulated during dehydration and rehydration compared to controls (Fig. 3d, e and Supplementary Data 5). Interestingly, only 6 proteins were differentially expressed in rehydration when compared to dehydration (Supplementary Data 5). In the medulla, 211 DEPs were identified in dehydrated animals compared to controls. Eighty-one of these DEPs returned to control levels during rehydration while 130 remained differentially expressed in dehydration and rehydration compared to controls. 1965 different DEPs emerged during rehydration compared to controls (Fig. 3d, f and Supplementary Data 5). Further, we characterised 428 DEPs (Supplementary Data 5) exclusive to the rehydration-dehydration comparison. Similar to the patterns showed by differentially expressed genes, the cortex exhibited more DEPs in dehydration than in rehydration compared to control while the opposite was found in the medulla.

**Lipid and cholesterol metabolic processes are also over-represented in the kidney proteomes.** Using ShinyGO[51] we also identified a significant enrichment ($p < 0.05$) of GO terms associated with cholesterol biosynthesis in both cortex and medulla at the protein level (Fig. 3g, h). In this case, we could not detect enrichment of GO terms related to ion transmembrane transport possibly due to the limitations of the protein extraction method which largely captures cytosolic peptides. However, other interesting biological processes and molecular functions were enriched such as "Exocytosis" (GO:0006887), "Membrane organization" (GO:0061024), "Regulation of cellular response to heat" (GO:0034650), "Cellular response to stress" (GO:0033554), "RNA splicing" (GO:0008380), "Heat shock protein binding" (GO:0031072) and "Unfolded protein binding" (GO:0051082) (Fig. 3g, h and Supplementary Data 6). All these functional groups may be of interest in the one-humped Arabian camel as potentially being involved in coping with environmental stressors. For instance, exocytosis of AQP2 is critical for water conservation, several ion transporters are rearranged in the membrane, and heat stress due to high ambient temperatures may lead to protein denaturation.

**Analyses of the transcriptomes-proteomes overlap further support a role for cholesterol during severe dehydration.** As documented above, variations in the hydration status of the dromedary camel elicited a profound response in the kidney at both transcript and protein levels. In total, we identified 1282 genes to be differentially expressed in at least one tissue and one condition. Similarly, we catalogued 3002 proteins with

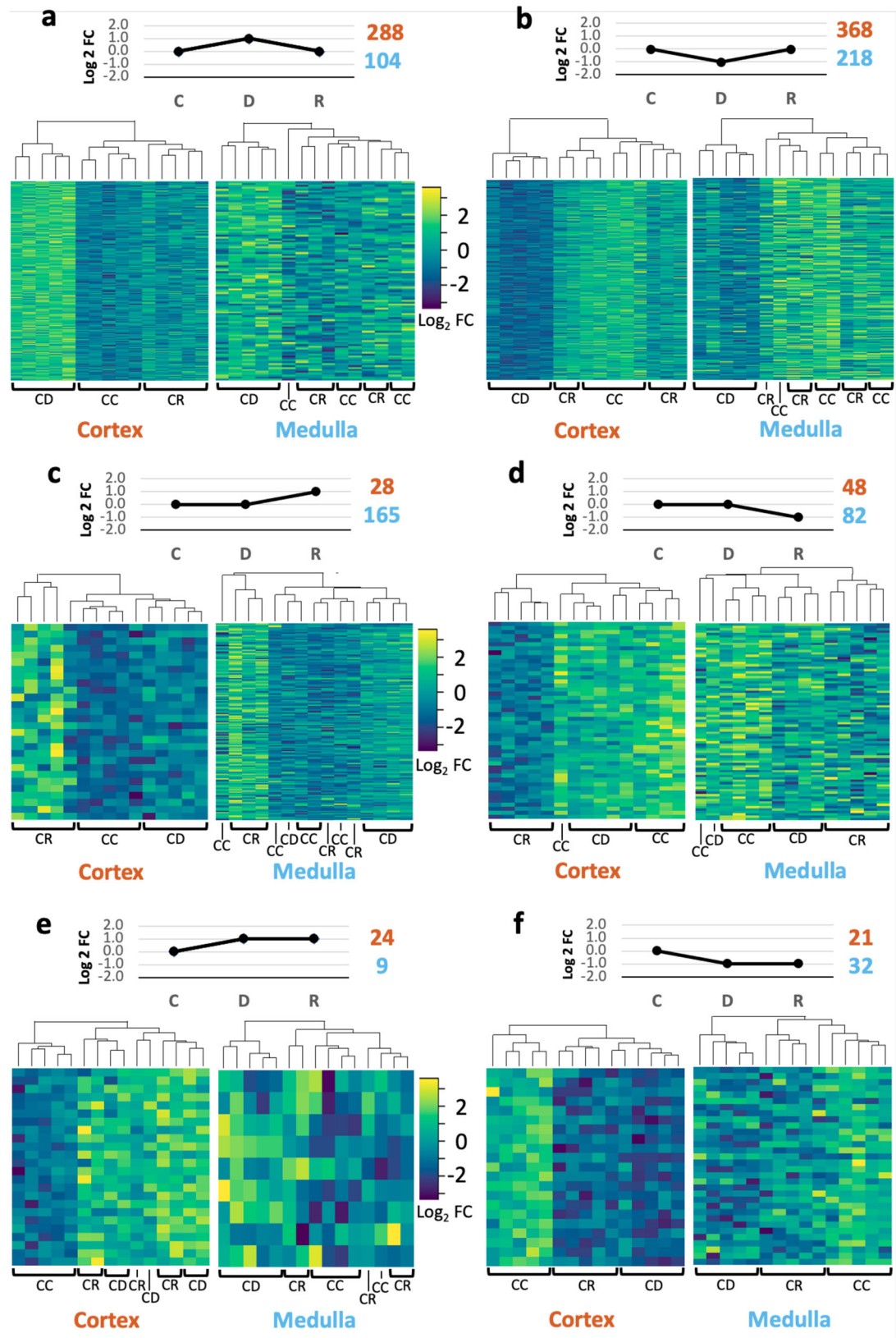

significantly changed peptide abundance. Interestingly, further comparisons between RNAseq and proteomic data revealed that only 157 DEGs had their corresponding protein differentially expressed. In the cortex, 92 and 10 genes were differentially expressed at both the transcript and the protein level during dehydration and rehydration, respectively (Fig. 4a, b and

Supplementary Data 7). Similarly, in the medulla, 19 and 50 genes were differentially expressed at both the transcript and the protein level during dehydration and rehydration, respectively (Fig. 4c, d and Supplementary Data 7). While transcriptomic analyses are more sensitive, only highly expressed genes were detectable at protein level by mass spectrometry (Supplementary

**Fig. 2 Expression profiles displayed by DEGs in the kidney of the one-humped Arabian camel during severe dehydration and acute rehydration. a–f** At the top of each panel, on the line chart, the expression profile displayed by the differentially expressed genes (DEGs) included in the heatmaps below (note that Log₂FC values are arbitrary and for visualization purposes only). The numbers of DEGs displaying the associated expression profile are shown for the cortex (orange) and the medulla (blue) on the top-right corner of each panel. Heatmaps show clustering of differentially expressed genes by condition. DEGs included in the heatmaps were filtered using a threshold of $p < 0.01$. We considered appropriate using a more stringent threshold for these analyses because we expected genes that follow a particular pattern to be more tightly regulated and therefore have smaller $p_{adj}$ values. C control, D dehydration, R rehydration, CC cortex control, CD cortex dehydration, CR cortex rehydration, MC medulla control, MD medulla dehydration, MR medulla rehydration.

Table ST1). Moreover, proteins that are immediately trafficked to the membrane or outside the cell were likely missed as the protein extraction method captures mostly the cytosolic fraction of the proteome since centrifugation discards membrane-bound proteins. Nevertheless, GO analyses of genes detected at both mRNA and protein levels during dehydration compared to controls showed, as previously described in the transcriptomes and proteomes separately, an enrichment of the "cholesterol biosynthetic process" (GO:0006695) and other GO categories related to lipid metabolism in the cortex (Fig. 4e and Supplementary Data 8) and the medulla (Fig. 4f and Supplementary Data 8).

**The cholesterol biosynthetic process is suppressed in the kidney of the one-humped Arabian camel during severe dehydration.** The RNAseq data suggested that 14 key genes involved in the cholesterol biosynthesis pathway (Table 1) were significantly ($p_{adj} < 0.05$) downregulated in the kidney medulla while 10 out of these 14 genes (excluding *ACAT2*, *FDFT1*, *MVD* and *PMVK*) were significantly ($p_{adj} < 0.05$) downregulated in the kidney cortex. During rehydration, expression of all these genes showed no significant difference compared to controls except for *ACAT2* and *DHCR24*, which remained significantly ($p_{adj} < 0.05$) downregulated in the medulla. However, 12 (*ACAT2*, *CYP51A1*, *DHCR24*, *DHCR7*, *FDPS*, *HMGCR*, *HMGCS1*, *LSS*, *MVD*, *MVK*, *NSDHL* and *SQLE*) and 7 (*DHCR24*, *DHCR7*, *FDPS*, *HMGCR*, *HMGCS1*, *LSS* and *SQLE*) genes were significantly ($p_{adj} < 0.05$) upregulated in medulla and cortex, respectively, during rehydration compared to dehydration. The remaining genes showed no change compared to dehydration. Thus, after an overall downregulation induced by dehydration, the expression of these genes returned, totally or partially, to control levels (Supplementary Table ST2).

Furthermore, we found a significant decrease ($p_{adj} < 0.05$) in peptide abundance for the proteins coded by 6 (*CYP51A1*, *DHCR7*, *FDPS*, *HMGCS1*, *LSS* and *NSDHL*) and 9 (*CYP51A1*, *DHCR7*, *FDPS*, *HMGCR*, *HMGCS1*, *LSS*, *NSDHL*, *PMVK* and *SQLE*) out of those 14 genes in the medulla and the cortex, respectively, during dehydration compared to controls. In the medulla; CYP51A1, DHCR7 and HMGCS1 were downregulated in dehydration compared to control and returned to control levels during rehydration while FDPS, LSS and NSDHL were significantly ($p_{adj} < 0.05$) downregulated in dehydrated animals compared to controls and remained downregulated during rehydration compared to control animals. HMGCR showed no significant changes across conditions. Interestingly, ACAT2 and PMVK did not change during dehydration compared to controls but were significantly ($p_{adj} < 0.05$) downregulated in rehydration compared to the other conditions. MVK protein abundance progressively decreased across treatments but only reached significant ($p_{adj} < 0.05$) levels in rehydration compared to controls. Lastly, SQLE protein abundance slightly increased in dehydration and was heavily suppressed during rehydration. Likewise, in the cortex, amongst the proteins significantly ($p_{adj} < 0.05$) downregulated during dehydration, CYP51A1, DHCR7, FDPS, LSS, NSDHL, PMVK and SQLE remained significantly

($p_{adj} < 0.05$) downregulated during rehydration compared to controls while HMGCR and HMGCS1 returned to control levels. ACAT2 and MVK showed no significant changes across conditions (Supplementary Table ST2). Unfortunately, DHCR24, FDFT1 and MVD were undetectable using mass spectrometry.

Based on our transcriptomic and proteomic data, we hypothesised that the suppression of cholesterol synthesis may play an important role in the kidney of the one-humped Arabian camel during water deprivation.

**RT-qPCR validation of genes encoding enzymes involved in cholesterol biosynthesis pathway.** We validated our RNAseq data using real-time quantitative PCR (RT-qPCR). In the medulla, $\Delta\Delta C_t$ fold changes ranged between 1.01–2.03, 0.20–0.87 and 0.44–1.75 in control, dehydrated and rehydrated animals, respectively. Relative mRNA expression dropped 0.37–1.58-fold in dehydrated animals compared to controls and increased up to 3.03-fold after rehydration compared to dehydrated animals. *ACAT2* ($p_{adj} = 0.014$), *CYP51A1* ($p_{adj} = 0.024$), *DHCR24* ($p_{adj} = 0.000$), *DHCR7* ($p_{adj} = 0.000$), *HMGCR* ($p_{adj} = 0.001$), *HMGCS1* ($p_{adj} = 0.005$), *LSS* ($p_{adj} = 0.001$), *MVD* ($p_{adj} = 0.015$), *NSDHL* ($p_{adj} = 0.013$), *PMVK* ($p_{adj} = 0.029$) and *SQLE* ($p_{adj} = 0.000$) were significantly downregulated during dehydration compared to controls. Their expression levels returned to control values during rehydration except for *ACAT2* ($p_{adj} = 0.044$), *DHCR24* ($p_{adj} = 0.046$), *NSDHL* ($p_{adj} = 0.012$) and *PMVK* ($p_{adj} = 0.033$), which remained downregulated compared to controls, and *HMGCR*, which was significantly upregulated ($p_{adj} = 0.004$) during rehydration compared to controls. *FDFT1* and *MVK* trended towards downregulation during dehydration, but changes were not significant. *FDFT1* was significantly ($p_{adj} = 0.002$) upregulated during rehydration compared to dehydration while *MVK* expression remained at dehydration levels. *FDPS* expression level remained unchanged throughout the experimental period in this tissue, which is the opposite of the trend shown by the rest of the genes (Fig. 5 and Supplementary Table ST3). These results confirm that the expression of cholesterol biosynthesis genes was overall suppressed during dehydration in the medulla of the Arabian dromedary kidney.

In the cortex, $\Delta\Delta C_t$ fold changes ranged between 1.01–2.51, 0.24–2.04 and 0.66–5.07 in control, dehydrated and rehydrated animals, respectively. Relative mRNA expression dropped 0.21–1.07-fold in dehydrated animals compared to controls and increased 0.33–3.03-fold after rehydration compared to dehydrated animals. *DHCR24* ($p_{adj} = 0.003$), *DHCR7* ($p_{adj} = 0.007$), *HMGCR* ($p_{adj} = 0.000$), *LSS* ($p_{adj} = 0.001$) and *SQLE* ($p_{adj} = 0.000$) were significantly downregulated during dehydration compared to controls, and expression levels returned to baseline after rehydration (Fig. 5 and Supplementary Table ST3).

It is important to note that *HMGCR*, the major rate-limiting enzyme in cholesterol synthesis[52–55], was significantly ($p_{adj} < 0.001$) downregulated in both cortex and medulla during dehydration compared to controls and returned to control expression levels, or above these, during rehydration.

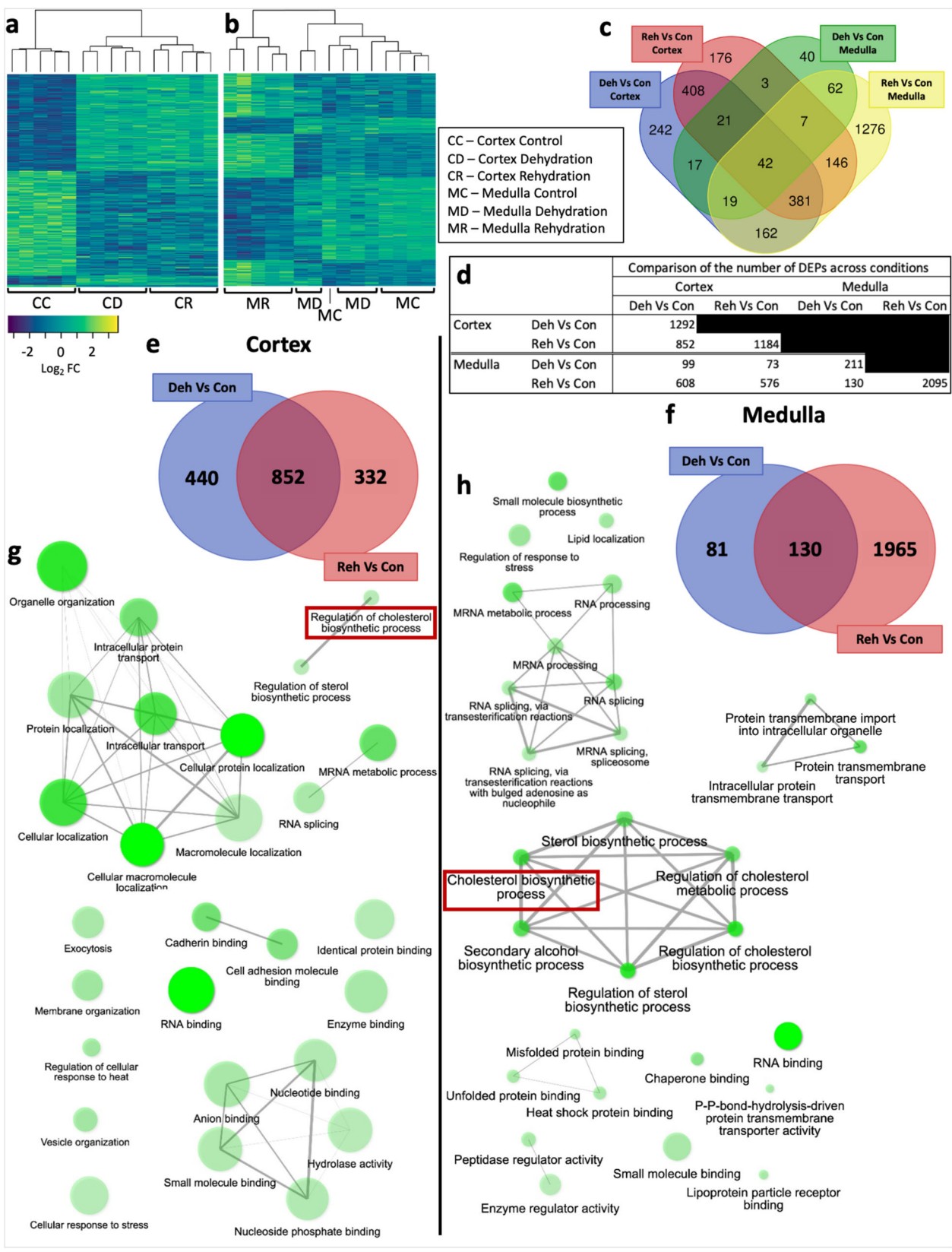

**Cholesterol analysis**. Our transcriptomic and proteomic data suggested that changes in the expression of enzymes involved in cholesterol biosynthesis might result in changes in cholesterol levels in the kidney as a consequence of dehydration. We therefore measured the cholesterol content in the kidney cortex and medulla. Despite excellent calibration and a CV% of 12% for the repeats, which indicated

good reproducibility, we did not find significant differences between the cortex and the medulla nor across conditions. Cholesterol levels were similar in both tissues, with $1.00 \pm 0.21$ and $0.90 \pm 0.19$ µg/mg ($n = 15$) of cholesterol measured in cortex and medulla, respectively. In the cortex, cholesterol content was $1.16 \pm 0.28$, $0.92 \pm 0.18$ and $0.92 \pm 0.18$ µg/mg ($n = 5$) in control, dehydrated and rehydrated

**Fig. 3 Analysis of the proteomes of the one-humped Arabian camel kidney during severe dehydration and acute rehydration. a, b** Heatmaps including all differentially expressed proteins (DEPs) identified in the cortex (**a**) and the medulla (**b**) across conditions show clustering of differentially expressed proteins by condition. **c** Venn plot displaying the overlap of differentially expressed proteins identified in dehydration and rehydration compared to control in cortex and medulla. **d** Absolute number of DEPs displayed in panels (**d**), (**e**) and (**f**). **e, f** Venn plots showing the distribution of DEPs during dehydration and rehydration compared to controls in the kidney cortex (**e**) and medulla (**f**) of the camel. Number of proteins differentially expressed solely in dehydration compared to control (blue), in rehydration compared to control (light red) and those differentially co-regulated in both conditions (dark red) are shown. **g, h** Gene Ontology (GO) networks displaying a subset of the most significantly enriched ($p < 0.05$) biological processes and molecular functions in the kidney cortex (**g**) and medulla (**h**) during dehydration. Two nodes are connected if they share 20% or more genes and thicker edges represent more overlapped genes, darker nodes are more significantly enriched gene sets and bigger nodes represent larger gene sets.

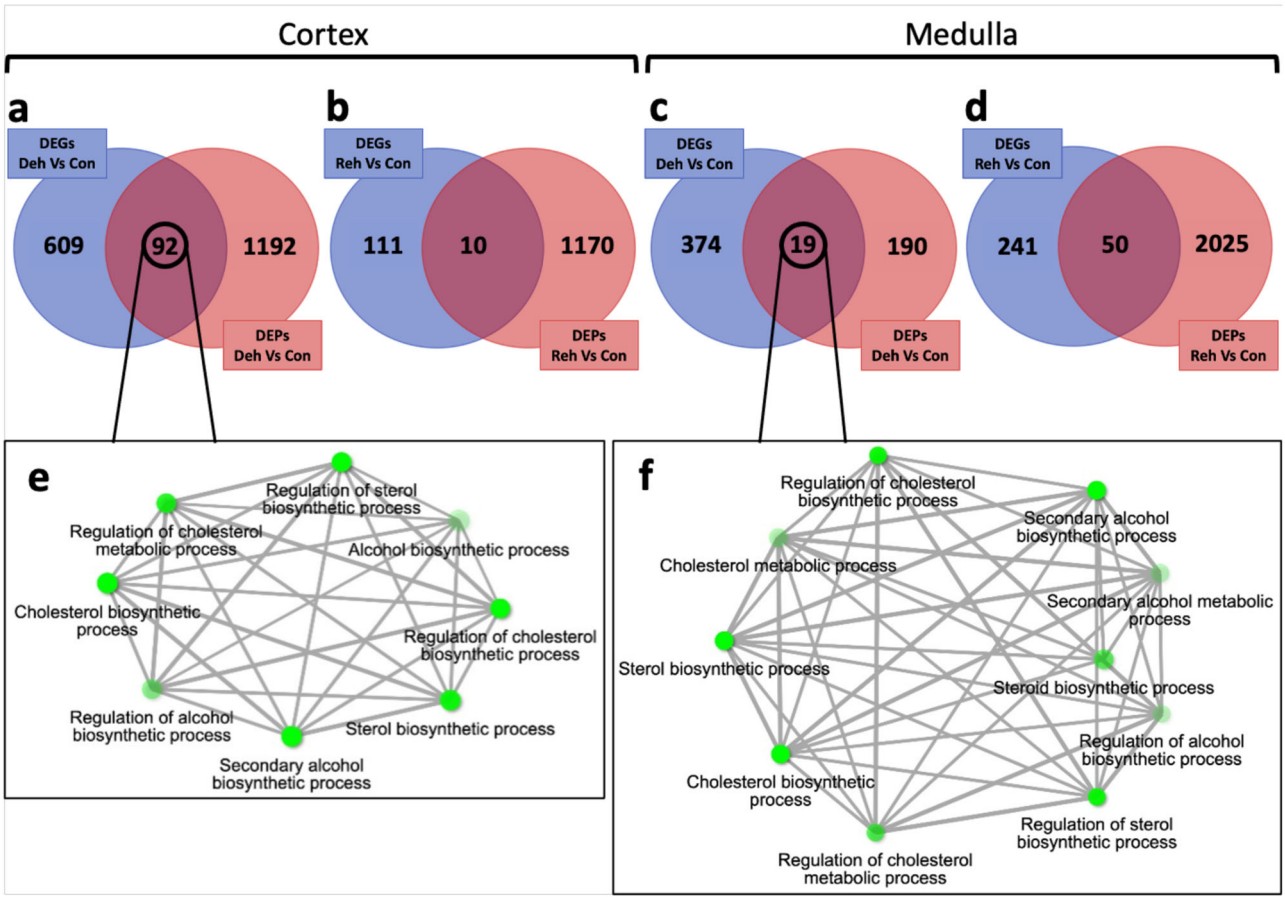

**Fig. 4 Tissue and treatment-specific comparison between matched DEGs and DEPs. a–d** Venn plots showing the overlap (dark red) of differentially expressed genes (DEGs) for which their corresponding differentially expressed proteins (DEPs) were also identified in the (**a**) cortex during dehydration, (**b**) cortex during rehydration, (**c**) medulla during dehydration and (**d**) medulla during rehydration. The Venn plots also display the number of genes (blue) and proteins (light red) differentially expressed at either RNA or protein level only. **e, f** Gene Ontology (GO) networks displaying a subset of the most significantly enriched ($p < 0.05$) biological processes in the kidney cortex (**e**) and medulla (**f**) during dehydration. Two nodes are connected if they share 20% or more genes and thicker edges represent more overlapped genes, darker nodes are more significantly enriched gene sets and bigger nodes represent larger gene sets.

animals, respectively. Likewise, in the medulla, cholesterol levels were $0.92 \pm 0.20$, $0.93 \pm 0.27$ and $0.87 \pm 0.11\,\mu g/mg$ ($n = 5$) in control, dehydrated and rehydrated animals, respectively (Fig. 6a). Given the analyses were carried out on unperfused tissue, we suspected the presence of blood would introduce considerable bias. This was confirmed by analysis of serum cholesterol, which revealed abundant cholesterol content. Further, these levels were significantly higher in dehydration ($715 \pm 155\,\mu g/ml$; $n = 6$; $p_{adj} = 0.004$) and remained elevated during rehydration ($548 \pm 116\,\mu g/ml$; $n = 6$; $p_{adj} = 0.086$) compared to control camels ($322 \pm 116\,\mu g/ml$; $n = 3$) (Fig. 6b).

According to our hypothesis, changes in the level of cholesterol should be more evident in the plasma membrane. Therefore, we quantified the amount of cholesterol in the plasma membrane

fraction isolated from kidney samples. Membrane cholesterol was reduced in dehydration compared to controls in both cortex and medulla, although variability prevented the differences reaching the significance threshold. In the cortex, the membrane cholesterol concentration was $418 \pm 127\,\mu g/mg$ of protein ($n = 3$) in control animals and decreased to $270 \pm 129\,\mu g/mg$ of protein ($n = 3$) in dehydrated animals. Similarly, in the medulla, membrane cholesterol concentration was $353 \pm 108\,\mu g/mg$ of protein ($n = 3$) and $256 \pm 40\,\mu g/mg$ of protein ($n = 3$) in control and dehydration, respectively (Fig. 6c). We further imaged kidney sections under widefield microscopy using the cholesterol-specific marker Filipin III and generated intensity surface plots to compare the intensity of the fluorescence signals. Signal was

**Table 1 Gene symbols, Ensembl ID's and names of the 14 enriched genes encoding enzymes involved in the cholesterol biosynthetic process (GO:0006695).**

| Gene symbols | Ensembl gene IDs | Gene name |
|---|---|---|
| ACAT2 | ENSCDRG00005015012 | Acetyl-CoA acetyltransferase |
| CYP51A1 | ENSCDRG00005022091 | Lanosterol 14-alpha demethylase |
| DHCR7 | ENSCDRG00005021350 | 7-Dehydrocholesterol reductase |
| DHCR24 | ENSCDRG00005012528 | 24-Dehydrocholesterol reductase |
| FDFT1 | ENSCDRG00005010723 | Farnesyl-diphosphate farnesyltransferase 1 |
| FDPS | ENSCDRG00005008269 | Farnesyl pyrophosphate synthase |
| HMGCR | ENSCDRG00005002491 | 3-hydroxy-3-methylglutaryl-CoA reductase |
| HMGCS1 | ENSCDRG00005014801 | Hydroxymethylglutaryl-CoA synthase |
| LSS | ENSCDRG00005021905 | Lanosterol synthase |
| MVD | ENSCDRG00005020571 | Diphosphomevalonate decarboxylase |
| MVK | ENSCDRG00005011437 | Mevalonate kinase |
| NSDHL | ENSCDRG00005017818 | Sterol-4-alpha-carboxylate 3-dehydrogenase |
| PMVK | ENSCDRG00005003806 | Phosphomevalonate kinase |
| SQLE | ENSCDRG00005004790 | Squalene monooxygenase |

reduced in dehydrated samples compared to controls in tubular cells of both tissues (Fig. 6d, e). Taken together, the results suggest that the suppression of the genes involved in cholesterol synthesis and the subsequent reduction in membrane cholesterol along the nephron of the kidney are, possibly, segment-specific.

Despite high variability that prevented some of the changes in gene expression assessed by RT-qPCR to reach the significance threshold, taken together, our RNAseq data, proteomes, RT-qPCR validation and membrane cholesterol quantification and visualization suggest that the suppression of genes involved in cholesterol biosynthesis and the subsequent reduction in membrane cholesterol are a global response in the kidney of the one-humped Arabian camel to dehydration, affecting both cortical and medullary segments.

**Genes coding for key ion and H$_2$O transporters are upregulated in the camel kidney during dehydration.** Based on our hypothesis of a role for cholesterol during dehydration, we identified differentially expressed genes with known roles in the countercurrent multiplication process in the kidney which were potentially affected by changes in the level of cholesterol in the cell/cell membrane. These genes are KCNJ8 (Potassium inwardly rectifier; ENSCDRG00005018658), SLC9A7 (Solute carrier NHE7; H$^+$/Na$^+$ antiporter; ENSCDRG00005020882), ATP1B3 (Na$^+$/K$^+$ - ATPase; ENSCDRG00005009339), and the gene coding for the water channel Aquaporin 2 (AQP2; ENSCDRG00005005216). KCNJ8 pumps K$^+$ ions from the cell into the lumen, SLC9A7 transports Na$^+$ ions into the cell, ATP1B3 actively pumps Na$^+$ into the interstitium and AQP2 allows water to flow through the cell into the interstitium to be reabsorbed. These genes were either significantly ($p_{adj} < 0.05$) upregulated in dehydration compared to controls or significantly ($p_{adj} < 0.05$) downregulated in rehydration compared to dehydration. Taking into account that no changes were identified between control and rehydrated animals, the expression level of these genes must be higher during dehydration compared to controls. SLC9A7 was selected for validation given that changes across conditions were in close proximity to the significance threshold (Supplementary Table ST2). We also identified other genes that were differentially expressed in at least one tissue/condition with known roles in the urine concentrating process. These genes were Aquaporin 1 (AQP1, ENSCDRG00005002776), Aquaporin 3 (AQP3, ENSCDRG00005020520) and Epithelial Sodium Channel (ENaC, ENSCDRG00005010881). These genes were validated using RT-qPCR. Although levels of significance were affected by high variability, KCNJ8 ($p_{adj} = 0.004$) and ATP1B3 (non-significant)

showed a ~0.5–2-fold increase in gene expression after dehydration in the medulla, while SLC9A7 expression level trended towards a smaller increase (non-significant). Relative expression of AQP2 showed a 2-fold increase ($p_{adj} = 0.000$). Expression levels for these genes returned to control rates after rehydration (Fig. 7a and Supplementary Table ST3). In the cortex, KCNJ8 ($p_{adj} = 0.003$), SLC9A7 ($p_{adj} = 0.001$) and ATP1B3 (non-significant) also showed a ~0.5–2-fold increase in gene expression after dehydration while AQP2 showed a smaller, non-significant increase. These levels returned to approximately baseline after rehydration, except for AQP2, which was strongly downregulated after rehydration compared to controls ($p_{adj} = 0.014$) and dehydration ($p_{adj} = 0.003$) in the cortex (Fig. 7b and Supplementary Table ST3). Furthermore, in the medulla, AQP1 and ENaC showed no significant changes across conditions, although ENaC seems to be less abundant in dehydrated camels. AQP3 was differentially upregulated (Log Fold change = ~2.9; $p_{adj} = 0.015$) in dehydration compared to controls and downregulated ($p_{adj} = 0.008$) in rehydration compared to dehydration. In the cortex, AQP1 (Log Fold change = ~2.1; $p_{adj} = 0.049$) and AQP3 (Log Fold change = ~2.7; $p_{adj} = 0.001$) were significantly upregulated in dehydration compared to control. Their expression returned to control levels during rehydration. ENaC remained unchanged across conditions in the cortex (Fig. 7).

We further compared our RNAseq data with two of the most complete and up to date reviews of the urinary concentrating mechanism[37,56] in order to elucidate whether other well-characterised ion/water transporters were differentially expressed in the Arabian camel during chronic dehydration and/or acute rehydration but did not reach the significance threshold (Table 2).

## Discussion

A plethora of adaptations to life in the desert have been described in desert-dwelling species. Recently, molecular and genomic approaches have started to unravel the underlying mechanisms of these adaptations[17,57–59]. The present study has revealed remarkable changes in the abundance of specific RNAs and proteins in the kidney of the one-humped Arabian camel during severe dehydration and subsequent acute rehydration. Gene Ontology analyses of dehydrated DEGs revealed an enrichment of the "cholesterol biosynthetic process" (GO:0008610) and the RNAseq and the proteomic data confirmed that key genes involved in the cholesterol biosynthesis pathway were overall downregulated. We further identified an enrichment of terms associated to "transmembrane transporter activity" (GO:0022857) in the kidney medulla. We confirmed that the expression of

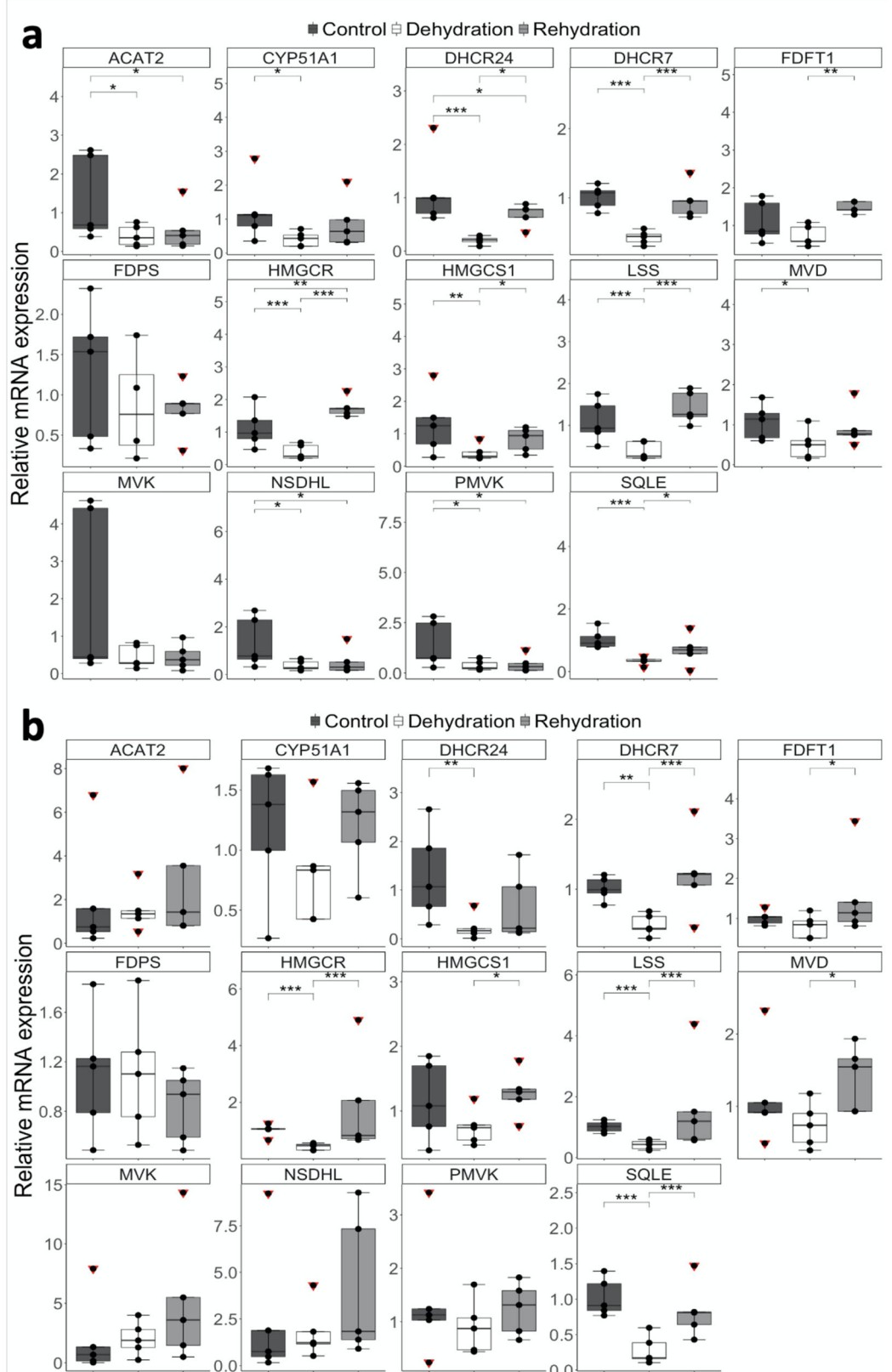

**Fig. 5 RT-qPCR validation of genes coding for enzymes involved in the cholesterol biosynthetic process.** Relative changes in gene expression of genes involved in cholesterol synthesis in the Arabian camel kidney **a** medulla and **b** cortex after severe dehydration and acute rehydration compared to controls. Comparison of the means by one-way ANOVA (Turkey's post hoc correction). The boxplots are presented with the S.E.M. ($n = 5$), centre lines show median, box edges delineate 25th and 75th percentiles and bars extend to minimum and maximum values. Individual data points represent biologically independent samples and data points within red triangles denote outliers, all the outliers highlighted were included for the statistical analyses. ***$p_{adj} <$ 0.001, **$p_{adj} < 0.01$, *$p_{adj} < 0.05$.

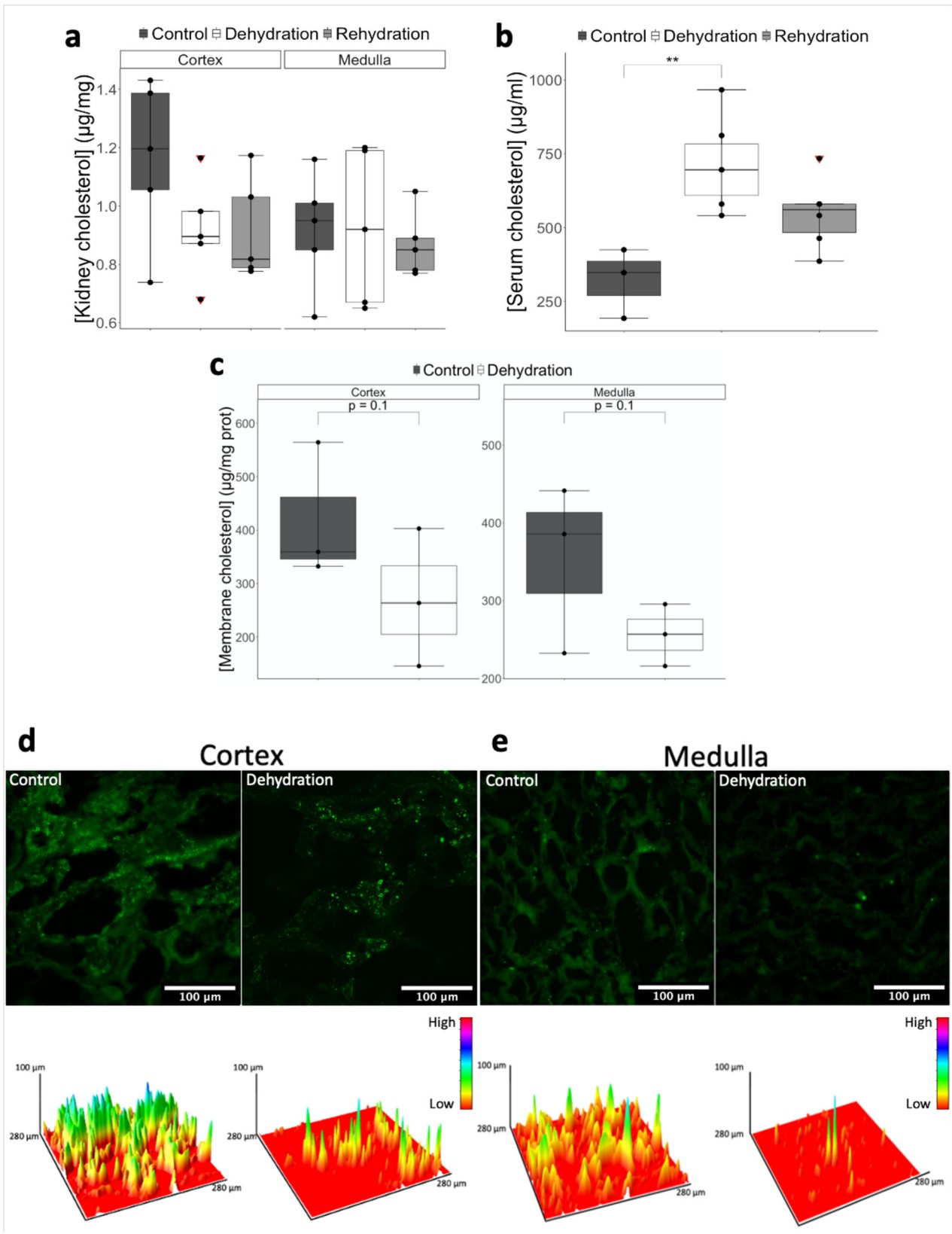

cholesterol biosynthesis genes was overall strongly suppressed in the cortex and the medulla using RT-qPCR, and further demonstrated that membrane cholesterol is reduced in the dromedary kidney during dehydration compared to controls. Taken together, our data suggest that the suppression of genes involved in cholesterol biosynthesis and the subsequent reduction in

membrane cholesterol are a global response in the kidney of the one-humped Arabian camel to dehydration, affecting both cortical and medullary segments. Based on our hypothesis of a role for cholesterol during dehydration, we identified DEGs with known roles in the urinary concentrating process in the kidneys which were potentially affected by changes in the level of

**Fig. 6 Cholesterol quantification in kidney cortex and medulla.** Cholesterol concentration in **a** kidney tissue, **b** blood serum and **c** kidney plasma membranes of the one-humped Arabian camel during control, dehydration and rehydration. Comparison of the means by one-way ANOVA (Turkey's post hoc correction). The boxplots are presented with the S.E.M. (kidney tissue, $n = 5$; serum cholesterol, $n = 3$ (control) and 6 (dehydration/rehydration); membrane cholesterol, $n = 3$), centre lines show median, box edges delineate 25th and 75th percentiles and bars extend to minimum and maximum values. Individual data points represent biologically independent samples and data points within red triangles denote outliers, all the outliers highlighted were included for the statistical analyses. ***$p_{adj} < 0.001$, **$p_{adj} < 0.01$, *$p_{adj} < 0.05$. **d, e** Widefield microscopy images of kidney sections stained with the cholesterol-specific, autofluorescent marker Filipin III from **d** cortex and **e** medulla of control and dehydrated camels. Cholesterol is shown in green. Scale bar: 100 μm. Below each microscopy image, the corresponding intensity surface plot shows the amount of fluorescence signal picked for each image (lower intensities are shown in the orange range while increasing intensity tends towards the pink range).

cholesterol in the cell/cell membrane. Thus, we further validated 4 genes coding for ion transporting proteins, namely *KCNJ8*, *SLC9A7*, *ATP1B3* and *ENaC*, and the genes coding for the water channels Aquaporin 1 (*AQP1*), Aquaporin 2 (*AQP2*) and Aquaporin 3 (*AQP3*). These genes showed overall upregulation during dehydration, except for *ENaC*.

Adaptive evolution occurs when advantageous features increase in population frequency in subsequent generations. Dehydration and excessive heat result in electrolyte imbalances. The major consequences are cell cycle arrest[60], DNA degradation[61], disruption of repair mechanisms[62], oxidative stress[63,64], disruption of transcription and translation[65] and inhibition of protein folding[66] which would ultimately result in cell death. Wu et al[17] showed that rapidly evolving GO categories identified in genomics studies of the kidney of the Bactrian camel were associated with "DNA damage and repair" (GO:0006974, GO:0003684, GO:0006302) and "apoptosis" (GO:0006917, GO:0043066) as well as the regulation of "body fluid levels" (GO:0050878). Further, MacManes[67] also showed upregulation of genes associated with both positive and negative regulation of apoptosis in the kidney of the Cactus mouse (*Peromyscus eremicus*), and argued that negative regulation of apoptosis may mitigate the effects of dehydration while prolonged stress and failure of survival mechanisms invoke apoptotic pathways. Therefore, developing mechanisms to avoid severe dehydration and prioritise water preservation is likely to be favoured in desert-dwelling animals, such as the one-humped Arabian camel, from an evolutionary point of view.

Previous research in the desert-adapted Cactus mouse has also revealed a significant downregulation of the cholesterol biosynthesis pathway. The author[67] describes GO analysis of the 213 identified downregulated genes and suggests that acute dehydration is linked to lower expression of genes related to cholesterol biosynthesis. In this case, RNAseq data showed that *DHCR7* ($p < 0.01$; logFC = 0.56), *FDFT1* ($p < 0.001$; logFC = 0.77), *FDPS* ($p < 0.0001$; logFC = 0.0.92), *HMGCR* ($p < 0.01$; logFC = 0.58), *HMGCS1* ($p < 0.01$; logFC = 0.79), *LSS* ($p < 0.0001$; logFC = 1.08), *MVD* ($p < 0.01$; logFC = 0.70), *NSDHL* ($p < 0.0001$; logFC = 1.18) and *SQLE* ($p < 0.0001$; logFC = 1.41) were significantly downregulated in dehydrated animals, however, no validation was performed. Despite transcriptomic analysis having been carried out on other relevant species, such as the Bactrian camel[17], the Olive mouse (*Abrothrix olivacea*)[58] and the Banner-tailed kangaroo rat (*Dipodomys spectabilis*)[68], data regarding the genes involved in the cholesterol biosynthesis pathway is, unfortunately, not publicly available. Further, recent attempts[69,70] to study the transcriptomes of the Bactrian camel under severe dehydration yielded very limited results in terms of differentially expressed genes, thus preventing potential cross-comparisons. However, an enrichment of rapidly evolving genes involved in lipid/cholesterol metabolism has repeatedly been described in the literature[17,59,67,71,72]. Together, the data suggest that cholesterol metabolism may play a role during dehydration in the kidney of a variety of desert-adapted species. Further, it is reasonable to argue

that it is yet another convergent evolutionary adaptation to cope with severe dehydration and its consequences, as it may play a role in facilitating the production of highly concentrated urine. However, variations in wild populations are likely so further research will be necessary to elucidate whether suppression of gene expression of genes involved in cholesterol biosynthesis is a widespread feature in desert-adapted mammals.

As our transcriptome data predicted changes in cholesterol content as a consequence of dehydration, we directly measured cholesterol in cortex and medulla of control, dehydrated and rehydrated kidneys. However, perfusing the kidneys was technically impossible during dissection, so blood was present in the tissue samples. Under normal conditions, plasma cholesterol in one-humped Arabian camels has been reported to be ~400 μg/ml (calculated from Mohri et al.[73]), and it was more than 30 times higher than that in male Bactrian camels according to Omidi et al.[74], which would strongly affect the signal to noise ratio making it difficult to detect any inherent differences in the tissue. We confirmed this fact by later analyses of serum cholesterol showing values as high as 800 μg/ml. Thus, assuming that plasma membrane cholesterol would be the primary target according to our hypothesis, we quantified and visualised membrane cholesterol in the kidney of one-humped Arabian camels and confirmed that it is reduced during dehydration.

Several ion channels and transporters, including members of all major families, are regulated by changes in the level of cholesterol in the cell. The most common effect is suppression of channel activity by an increase in membrane cholesterol[75]. In general, three mechanisms have been proposed to account for the sensitivity of ion channels to cholesterol: (1) direct binding of a cholesterol molecule to the channel protein[76], (2) an increase in the energetic cost of channel transitions between closed and open states due to an increase in the stiffness of lipid membrane bilayer[77–79] and (3) association of the channels with other regulatory proteins[80,81]. Most of the channels studied seem to be suppressed by increasing levels of cholesterol, although some channel families are lipid raft-dependent and/or more active when levels of sterols are above normal[75,82–84]. It is also speculated that lipid depletion in the brain associated with aging and/or disease may lead to alterations in channel function[85]. Furthermore, it has been hypothesised that variations in cholesterol content in the neuron cell membrane can indirectly modulate $Ca^+$ permeability in Alzheimer's patients[85,86].

Inwardly rectifying $K^+$ channels are strongly affected by the levels of cholesterol. Although mostly investigated in Kir.2[82,84,87,88] there is also evidence for Kir.6[88,89], which is an alias for *KCNJ8*. Apparently, cholesterol affects the state of the ion channel by directly binding to the CD loop of the C-Terminus, which changes such state from active-inactive to silent[82]. We show a significant increase in the expression of *KCNJ8* in our RNAseq data, which was further validated by RT-qPCR, in both cortex and medulla after severe dehydration which is consistent with the results previously shown in Bactrian camel[17] and Olive mouse[58,90]. $K^+$ inwardly rectifiers are localised in the TAL and

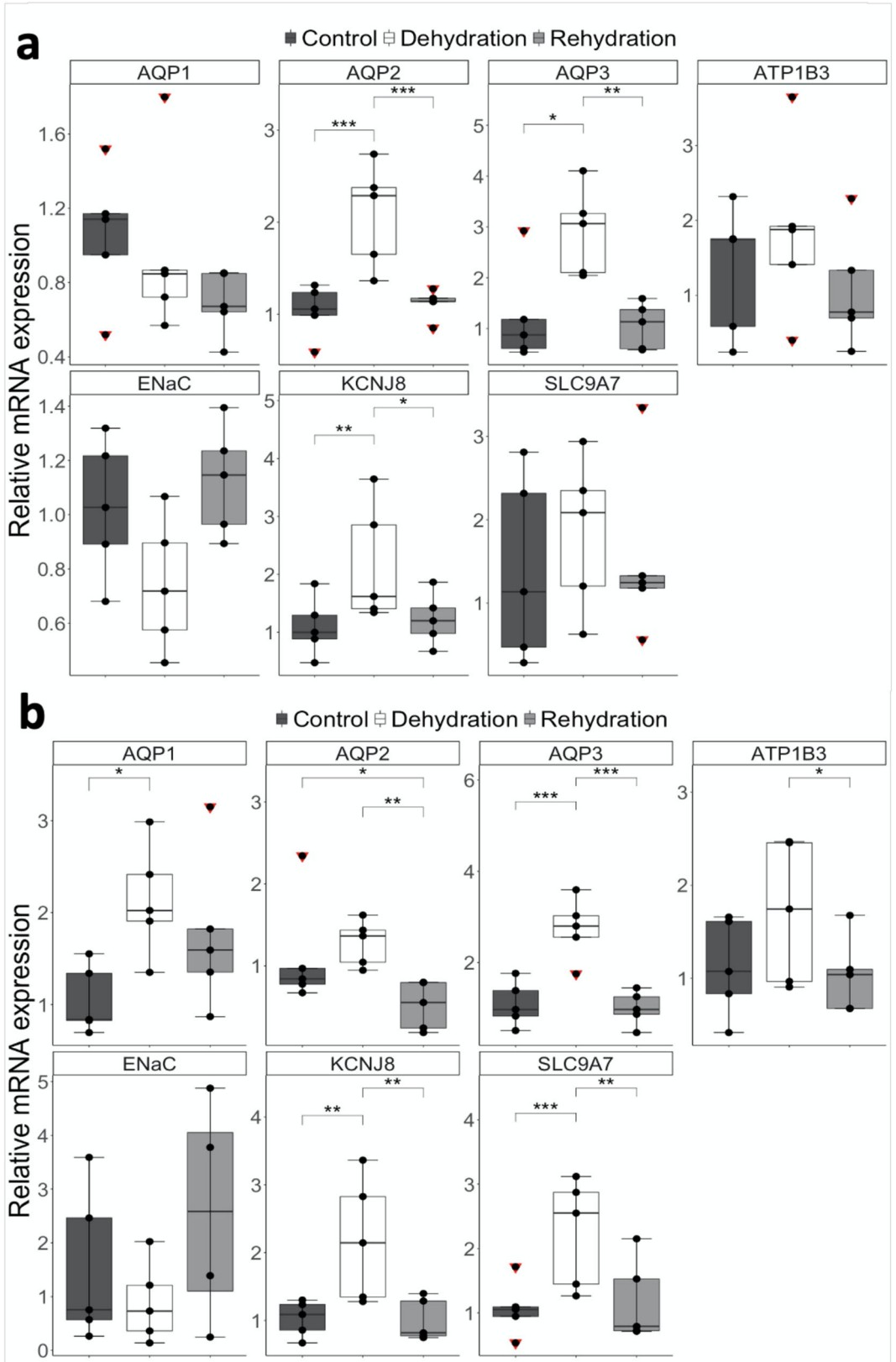

**Fig. 7 RT-qPCR validation of genes coding for ion transporters and aquaporins.** Relative changes in gene expression of key genes involved in solute and water transport in the Arabian camel kidney **a** medulla and **b** cortex after severe dehydration and acute rehydration compared to controls. Comparison of the means by one-way ANOVA (Turkey's post hoc correction). The boxplots are presented with the S.E.M. ($n = 5$), centre lines show median, box edges delineate 25th and 75th percentiles and bars extend to minimum and maximum values. Individual data points represent biologically independent samples and data points within red triangles denote outliers, all the outliers highlighted were included for the statistical analyses. ***$p_{adj} < 0.001$, **$p_{adj} < 0.01$, *$p_{adj} < 0.05$.

**Table 2 Other ion transporters and aquaporins with roles in the kidney.**

| Gene name | Gene symbols | Arabian camel | | | | | | Fenton and Knepper | | Sands and Layton |
| --- | --- | --- | --- | --- | --- | --- | --- | --- | --- | --- |
| | | Cortex | | | Medulla | | | Cortex | Medulla | Medulla |
| | | Deh Vs Con | Reh Vs Con | Reh Vs Deh | Deh Vs Con | Reh Vs Con | Reh Vs Deh | | | |
| Aquaporin 4 | AQP4 | ND | ND | ND | ND | ND | ND | | | X |
| Chloride channel | CLC-K2 | ND | ND | ND | ND | ND | ND | X | X | |
| Chloride channel | CLC-K1 | ND | ND | ND | ND | ND | ND | | | X |
| K+-Cl− cotransporter | KCC4 | = | = | = | = | = | = | X | X | |
| Na+-Cl− cotransporter | NCC/Slc12a3 | = | = | = | = | = | = | X | | |
| Na+-H+ exchanger | NHE3/Slc9a3 | = | = | = | = | = | = | X | X | |
| Na+-K+-2Cl− cotransporter | NKCC2 | = | = | = | = | = | = | | | X |

Comparison between our RNAseq data and two of the most complete reviews[37,56] of the urinary concentrating mechanisms of additional solute carriers and aquaporins with roles in the urine concentration process. The table shows transporters that have been previously described as important for the ability of the mammalian kidney to produce highly concentrated urine which are not differentially expressed in the one-humped Arabian camel. =, no significant change; ND, not detected, X, mentioned previously in either or both reviews.

the cortical collecting duct (CCD) of the nephron[37]. In the TAL, they exert a fundamental task in pumping $K^+$ ions from the cell into the lumen of the tubule as $Na^+$ ions are continuously pumped in reverse fashion during the countercurrent multiplication process, thus, maintaining a relatively constant cytosolic concentration. In the CCD, these $K^+$ channels regulate urinary potassium excretion and systemic balance. Both processes are regulated by the neuropeptide hormone arginine vasopressin (AVP)[91,92], well-known to be secreted into the blood during dehydration in mammals[37,93].

Moreover, it is now clear that $Na^+/H^+$ exchangers and $Na^+$-$K^+$−$2Cl^-$ cotransporters are major $Na^+$ absorptive pathways while NCC ($Na^+$-$Cl^-$ symporter) and ENaC (Sodium Channel Epithelial 1 Subunit Alpha) support their function in the distal nephron[37,41,42]. The major source of passive $Na^+$ diffusion into cells is thought to arise from the $Na^+/H^+$ antiporters, such as the one coded by SLC9A7[37,94,95], and the $Na^+$-$K^+$−$2Cl^-$ cotransporter coded by NKCC2[41,96]. The former utilizes the energy stored in the $Na^+$ concentration gradient to actively extrude $H^+$ ions out of the cell and prevent the environment of the cytoplasm from becoming too acidic[97]. Furthermore, Haines[98] established that both hydrogen and sodium ions leak through the lipid bilayer and proposed that cells use certain lipids to inhibit cation leakage through the membrane bilayers. It is assumed that $Na^+$ leaks through the bilayer by a defect mechanism. For $Na^+$ leakage in animal plasma membranes, the evidence suggests that cholesterol is a key inhibitor of $Na^+$ leakage. Experiments have shown that it reduces $Na^+$ and $K^+$ leakage through lipid bilayers to approximately one-third of that in bilayers that lack the sterol. Moreover, cholesterol content of membranes has some correlation with extracellular $[Na^+]$. For example, in intestinal epithelial cells, cholesterol occurs in higher concentration at the basolateral membranes facing blood than it does at the brush border membranes. The reverse is observed in renal epithelial cells where the basolateral membrane is facing blood, but the apical membrane is facing a slightly higher concentration of sodium to be excreted with the urine. The tight junctions of these cells act as a barrier preventing lateral diffusion of cholesterol between membrane domains. This suggests that the concentration of cholesterol in membrane lipids may be partly regulated by the $Na^+$ concentration facing the membrane. Sodium ions also cross the tubule membrane via the paracellular pathway[99] which is probably facilitated by the reduction of membrane rigidity resulting from lower cholesterol levels. We found that SLC9A7 is significantly more expressed in the cortex during dehydration, where considerable amounts of $Na^+$ and water are reabsorbed. This is also the case in the medulla, although the differences are significant only in the RNAseq data and could not be validated as such by RT-qPCR. Expression levels returned to baseline after rehydration. Again, similar findings were shown in Bactrian camel[17] and Olive mouse[58,90], where members of the SLC9 family were significantly overexpressed in cortex and medulla after dehydration. Interestingly, $Na^+/H^+$ antiporters colocalise with $K^+$ inwardly rectifiers (KCNJ8) and $Na^+$-$K^+$-ATPase (ATP1B3) in the TAL of the Loop of Henle and with $Na^+$-$K^+$-ATPase (ATP1B3) in the inner medulla and proximal convoluted tubule[37].

Further, early studies by Yeagle et al.[100] showed that a 50–100% increase in the level of cholesterol inhibited $Na^+$-$K^+$-ATPase activity by 30–70%, although recent in vitro studies proved the opposite[101]. However, membrane cholesterol depletion is expected to increase membrane passive permeability to $Na^+$[98] which allows $Na^+$ to flow into the cell, increasing the cytoplasmatic $[Na^+]$. Because under physiological conditions cytosolic $Na^+$-stimulated phosphorylation by ATP is a major rate-determining of the enzymatic cycle, cholesterol depletion

should lead to Na$^+$-K$^+$-ATPase stimulation[97]. Thus, an indirect effect of cholesterol depletion seems to be important in the kidney during dehydration and whether or not cholesterol directly affects Na$^+$-K$^+$-ATPase activity may not be relevant. Wu et al.[17] showed a significant increase in the expression of genes coding for ATPases after dehydration in Bactrian camel kidney which was also seen in Olive mice living in dry environments (e.g., Patagonian steppe)[58,90]. We also show an increase, although not significant, in the expression of *ATP1B3* after severe dehydration in both cortex and medulla. A major role of the Na$^+$-K$^+$-ATPase in the kidney is to prevent accumulation of the Na$^+$ ions moving from the lumen of the Loop of Henle into the cell[41], which would otherwise lead to an osmotic swelling[97]. This is supported by the localization of the ATPases in the kidney, which are found in the basolateral membrane of the TAL and the CCD[37].

Lastly, *AQP1* was significantly upregulated in the cortex during dehydration which is in accordance with previous investigations. It is widely accepted that AQP1 localises in both the apical and the basolateral membranes of the proximal tubules where it plays an essential role by reabsorbing water across the tubular cells[37,102,103]. Conversely, it remained unchanged in the medulla despite previous research showing that it localises in the tDL of long Loops of Henle in both model organisms[37,102,104] and desert living species[103]. In the tDL, AQP1 exerts an important role in water reabsorption. As far as we know, there is no evidence of interaction between AQP1 and cholesterol. *AQP3* was significantly upregulated in both cortex and medulla during dehydration in Arabian camel. This is in agreement with previous investigations reporting expression of AQP3 in the basolateral membrane of the cortical[37], outer and inner[102,104–106] collecting ducts. At this location, AQP3 transports water from the cellular compartment into the interstitium, a process required to maximise the ability of the kidney to concentrate urine[107]. To the best of our knowledge, there is only one reference describing the interaction between AQP3 and cholesterol. The authors showed an increase in *Aqp3* expression in *DHCR24*$^{-/-}$ mice, which is an enzyme that converts desmosterol to cholesterol. However, they further concluded that *Aqp3* expression in the kidney was equivalent in wild-type and knock-out mice, thus, this interaction seems to be specific to the epidermis[108]. The effects of the level of cholesterol on AQP2 are better characterised. Upon activation of water conservation mechanisms, AQP2 appears to localise mostly in the apical and basolateral membranes of the medullary collecting ducts[37,109–113]. Under resting conditions, AQP2 is found primarily in intracellular vesicles beneath the apical membrane. In response to AVP binding to the V2 receptor at the basolateral membrane, AQP2 is phosphorylated, a crucial event for the exocytosis of AQP2-bearing vesicles at the apical plasma membrane[114–116]. This process is more active in dromedaries during the summer, presumably in preparation for the more challenging conditions of this season[117]. AQP2 redistribution to the apical membrane has been closely correlated with a dramatic increase in membrane water permeability of the, otherwise nearly impermeable, collecting duct and is responsible for the water reabsorption in this segment of the nephron. Withdrawal of AVP triggers the endocytosis of AQP2-containing vesicles and restores the water-impermeable state of the apical membrane[118]. Interestingly, cholesterol depletion results in the accumulation of AQP2 in the cell membrane, an effect similar to that induced by AVP and potentially independent of its action[119]. Also, mild cholesterol depletion promoted a 3-fold increase of AQP2 at the apical membrane in MCD4 cells. An endocytosis assay proved that cholesterol depletion affects AQP2 endocytosis, thus, increasing its steady-state abundance at the apical plasma membrane[118]. We detected a significant increase in the expression of *AQP2* in the medulla after dehydration while a similar

trend was detected in the cortex although not reaching the significance threshold due to the high inter-sample variability. Moreover, *AQP2* was within the top 3 most upregulated genes in the medulla and the cortex of the Bactrian camel[17] and *AQP2* gene expression was suppressed during acute rehydration at the same time as the expression of cholesterol biosynthesis genes returned to control levels. Thus, a reduction in AVP levels together with cholesterol repletion would restore AQP2 endocytosis. This is consistent with the results shown by Procino et al. and Lu et al.[118,119] in cholesterol depleted cell lines.

Nevertheless, some limitations of this study must be addressed. Despite best efforts to minimize the time between slaughter and tissue harvesting, some samples were exposed longer at room temperature before freezing due to the size of the animals and the complexity of the dissection, potentially affecting the quality of the RNA. Moreover, the samples were kept frozen for a number of years. Again, the samples were carefully stored at −80 °C assuring maximum quality, but some degradation might have occurred. Nevertheless, RNA quality measured by Nanodrop matched our usual conventional laboratory standards and RINs were well within acceptable standards (Supplementary Table ST4). In fact, we believe that the majority of the variability accrue to the animals themselves. We detected differences in weight, with some animals being clearly heavier than others, which could alter the degree of response to dehydration. Further, despite all camels being in the age range of 4–5 years, the small variation could influence the ability to respond to dehydration[120]. Physical differences were obvious as well, with animals differing, for example, in colour. While coat colour per se is not expected to affect differential gene expression in the kidney, it is very well documented in the literature (see 'Introduction') that intense crossbreeding to give rise to dromedaries with specific characteristics is a common practise in the region, and colour is very often used to determine which animals are crossbred to each other. This could have some effects at a genetic level.

Furthermore, studies based on non-model species are expected to have high biological variance which adds considerable uncertainty to gene expression measurements regardless of the technology applied. In order to mitigate biased estimations of gene expression, the Encyclopaedia of DNA Elements Consortium (ENCODE) guidelines recommend at least two biological replicates and 30 million paired-end reads per sample for studies of human RNAseq data. However, it has been shown that sequencing efforts in the range of 5–20 million mapped reads per sample provide sufficient depth to accurately quantify gene expression. Previous statistical research has shown that even a minor increase in sample size provides the greatest power for differential expression analysis from RNAseq data, and that was a key aim of our experimental design since DESeq2 depends on accurately modelling biological variance. When sample sizes are small (≤3 biological replicates per condition), this method suffers from high false-positive rates. In such cases, even a small increase in sample size can significantly improve experimental power and the accuracy of detecting DEGs (for review, see ref. [121]). We used Lamorte's power calculation to calculate the sample size assuming a minimum power of 0.8. In addition, we adhere to the general guidelines mentioned above and stay well above the recommended 5–20 million reads per sample and the minimum sample size.

However, we acknowledge that variability between some of the $\Delta\Delta C_t$ values is higher than we would have liked. The reason behind is the strong variability in gene expression between biological replicates which results in a wide range of $C_t$ values shown within each experimental group. Assuming low $C_t$ variability, average $\Delta\Delta C_t$ for control animals always trends to 1, however, large variability within this group results in elevated average $\Delta\Delta C_t$

values after log transformation which utterly affects calculations of differential gene expression. Anyway, this is consistent with the transcriptomic data, where large variation in the number of reads is also shown. In addition, caveats regarding the protein extraction protocol need to be addressed. Despite the protocol we used being routinely used, the speed of centrifugation eliminates most of the debris but inevitably precipitates a small subset of the protein fraction as well. On the other hand, RIPA buffer is widely utilised to solubilise proteins but, again, a fraction of them is insoluble in this buffer. Either of these subsets would not be detected by mass spectrometry, thus, we could not have analysed these proteins. Regardless of the above, later attempts to quantitatively assess differential gene expression in dehydrated and rehydrated animals from RT-qPCR results would lack reliability. However, it was not the intention of the authors to do so, but solely validate, using an independent method, our RNAseq data. Together, these challenges should be understood as an unavoidable limitation of doing research with non-model species and we are confident to say that the variability shown in these experiments is due to biological differences rather than technical bias and, therefore, we are confident with the results presented here.

In this study, we have comprehensively described the transcriptomes and proteomes of the one-humped Arabian camel kidney cortex and medulla during severe dehydration and subsequent acute rehydration. We then analysed differential gene and protein expression and observed an enrichment of genes involved in the cholesterol biosynthesis pathway. Our datasets suggested a suppression of the cholesterol biosynthesis pathway. Based on our hypothesis of a role for cholesterol during dehydration we further validated genes coding for enzymes involved in the cholesterol biosynthesis pathway and DEGs encoding ion and water transporting proteins and established potential interactions. Taking all data into account, we argue that ion/water transport in the kidney of dehydrated camels may be indirectly enhanced by the suppression of the cholesterol biosynthetic process, and the subsequent reduction in membrane cholesterol. Cholesterol depletion would increase the activity/abundance of important ion and water transporters throughout the kidney nephron, including the convoluted tubules and the cortical collecting ducts as well as segments of the medulla involved in generating the single effect in the TAL of the Loop of Henle[37], which facilitates the successive countercurrent multiplication process that takes place in the kidney (Fig. 8). A decrease in the level of membrane cholesterol would allow a larger diffusion of $Na^+$ ions from the tubule lumen into the cell, such increase in $Na^+$ concentration would be compensated by the increasing amount of $K^+$ inwardly rectifiers excreting $K^+$ ions into the lumen, thus maintaining relative osmotic pressure within the cell. Both processes benefit from some degree of cholesterol depletion. Lastly, the increased cytosolic $Na^+$ concentration would stimulate ATPase activity which would pump those $Na^+$ ions into the interstitium, creating the osmotic gradient widely described as the single effect (for review, see ref. [37]). Subsequently, water would be dragged following the osmotic gradient from the tDL and collecting ducts through, among others, the highly expressed AQP2, and reabsorbed through the vasa recta, consequently increasing urine concentration, a process well-known to be regulated by AVP[37,40,41,93,109,116]. Nevertheless, we acknowledge that further research, perhaps repeating similar studies after microdissecting the kidney nephron, will be necessary to precisely localise where these changes in the level of cholesterol are most prominent within the camel nephron.

## Methods

**Animals**. Nineteen male dromedary camels aged 4–5 years, body weight range 276–416 kg, were used in the present study. The camels were supplied with alfalfa hay as feed and ranch-housed in the United Arab Emirates. Veterinary supervision was provided throughout the experimental period and no signs of distress or illness were identified. After a short adaptive period, the camels were divided into three groups, control ($n = 5$), dehydrated ($n = 8$) and rehydrated ($n = 6$). The control group had free access to food and water during the entire experimental period. The dehydrated group was water-deprived but had ad libitum access to food for 20 days. Meanwhile, the rehydrated group was subjected to the same protocol as the dehydrated animals followed by ad libitum water supply for 3 additional days. After the experimental period, the camels were sacrificed in the local central abattoir for human consumption in April 2016. Kidney samples were harvested within 1 h after killing the animals, immediately frozen in liquid nitrogen and kept at −80 °C for later physiological, histological, morphological and molecular analysis. Samples were shipped frozen on dry ice to the University of Bristol under the auspices of a DEFRA Import Licence (TARP/2016/063). This study was approved by the Animal Ethics Committee of the United Arab Emirates University (approval ID: AE/15/38) and the University of Bristol Animal Welfare and Ethical Review Board.

**Transcriptomic analysis**. Total RNA was extracted from fresh frozen kidney medulla and cortex of 5 control, 5 dehydrated and 5 rehydrated camels (for a total of 30 samples). These 15 animals were randomly selected out of the 19 animals included in the experimental set up while the remaining 4 were used for preliminary analyses only. Frozen tissue was added to a pre-cooled mortar containing a small volume of liquid nitrogen. Tissue was then grounded to a fine powder using a pestle and the nitrogen was allowed to evaporate. The powdered tissue was placed into pre-chilled tubes on dry ice and stored at −80 °C. For RNA extraction, 1 ml of Qiazol (79306, Qiagen) was added to 100 mg of powdered tissue, then immediately vortexed for 2 min, then incubated on ice for 10 min. Lysed samples were spun at $12{,}000 \times g$ for 10 min at 4 °C and the supernatant was transferred to a new tube. 1/5 volume Chloroform (22711.244, VWR) was added to each sample and vortexed for 15 s. Samples were then spun at $12{,}000 \times g$ for 15 min at 4 °C. Three hundred and fifty microliters of the upper phase was removed and mixed with an equal volume of absolute ethanol. RNA was extracted using ZymoTM Direct-Zol RNA miniprep (Zymo Research) as per the manufacturer's instructions. Total RNA concentration and 260/280 ratios were measured using a Nanodrop 2000c (Thermo Scientific). Illumina Sequencing was performed by Bristol Genomics Facility, University of Bristol, using the poly-A selection method for library preparation and the Illumina NextSeq500 system. RIN were 8.1 ± 0.5, 7.7 ± 0.2 and 7.6 ± 0.6 for control, dehydrated and rehydrated cortex samples, and 8.5 ± 0.5, 8.8 ± 0.2 and 8.7 ± 0.4 for control, dehydrated and rehydrated medulla samples, respectively (Supplementary Table ST4). No statistical differences were found between conditions, so we assume no bias towards specific groups.

An in-house computer (Dell PowerEd geR820 12 core personal supercomputer equipped with 512GB RAM and 12x1TB HDD) was used to process sequencing data using a bespoke pipeline. First, reads were trimmed of adaptor sequences using BBDuk tool followed by FastQC[122] quality control. STAR[123] was used to map reads using default settings to the publicly available *Camelus dromedarius* genome assembly GCA_000803125.2 (CamDro2) downloaded from the Ensembl 100 database[124]. Mapped reads were summarised using FeatureCounts[125] grouping to gene identifiers. DESeq2[126] in Rv3.6.1[127] was used to estimate differential expression of genes between conditions.

**Proteomic analysis**. The same camel kidneys selected for transcriptomic analysis were used for proteomic analysis. Protein extraction was carried out by adding 10 µl of extraction buffer with protease inhibitors per mg of frozen grounded sample to lyse cells. Extraction buffer with protease inhibitors consisted of 0.1 µl protease inhibitor (P8340, Sigma, 100x in DMSO), 0.1 µl Phenylmethanefonyl Flouride (P7626, Sigma, 100x in EtOH), 1 µl phosphatase inhibitor (PhosSTOP 04906837001, Roche, 10x in RIPA), and 8.8 µl RIPA buffer. The mixture was vortexed at a high speed for 1 min then incubated on ice for 30 min. Lysed samples were spun at $12{,}000 \times g$ for 20 min at 4 °C to retain only proteins in the supernatant. Then supernatant was transferred to a fresh tube. Colorimetric-based Bradford assay was used to measure protein concentration[128,129].

Relative peptide abundance was measured using tandem mass tagging mass spectrometry (TMT-MS) performed at the Bristol Proteomics Facility, University of Bristol. TMT-MS is a quantitative proteomic approach which allows comparison of protein abundance between multiple samples in a single experiment. Briefly, samples were digested and labelled with a set of amine-specific isobaric tags to obtain differentially labelled peptides of identical mass. Samples were then pooled and fractionated by chromatography and later analysed by nano-LC MS/MS using a Orbitrap Fusion Tribrid mass spectrometer. Quantification of peptides by mass spectrometry in each sample was compared to a pooled sample total to give a peptide abundance ratio. Proteome Discoverer v2.1 software was used to filter the data with a 5% FDR cut off and remove any contaminants identified. Normalisation to total protein content of each sample was performed for the final protein abundance ration. Differential protein expression was calculated using the DEqMS R package[130] using log-transformed and median normalised protein abundance ratio values.

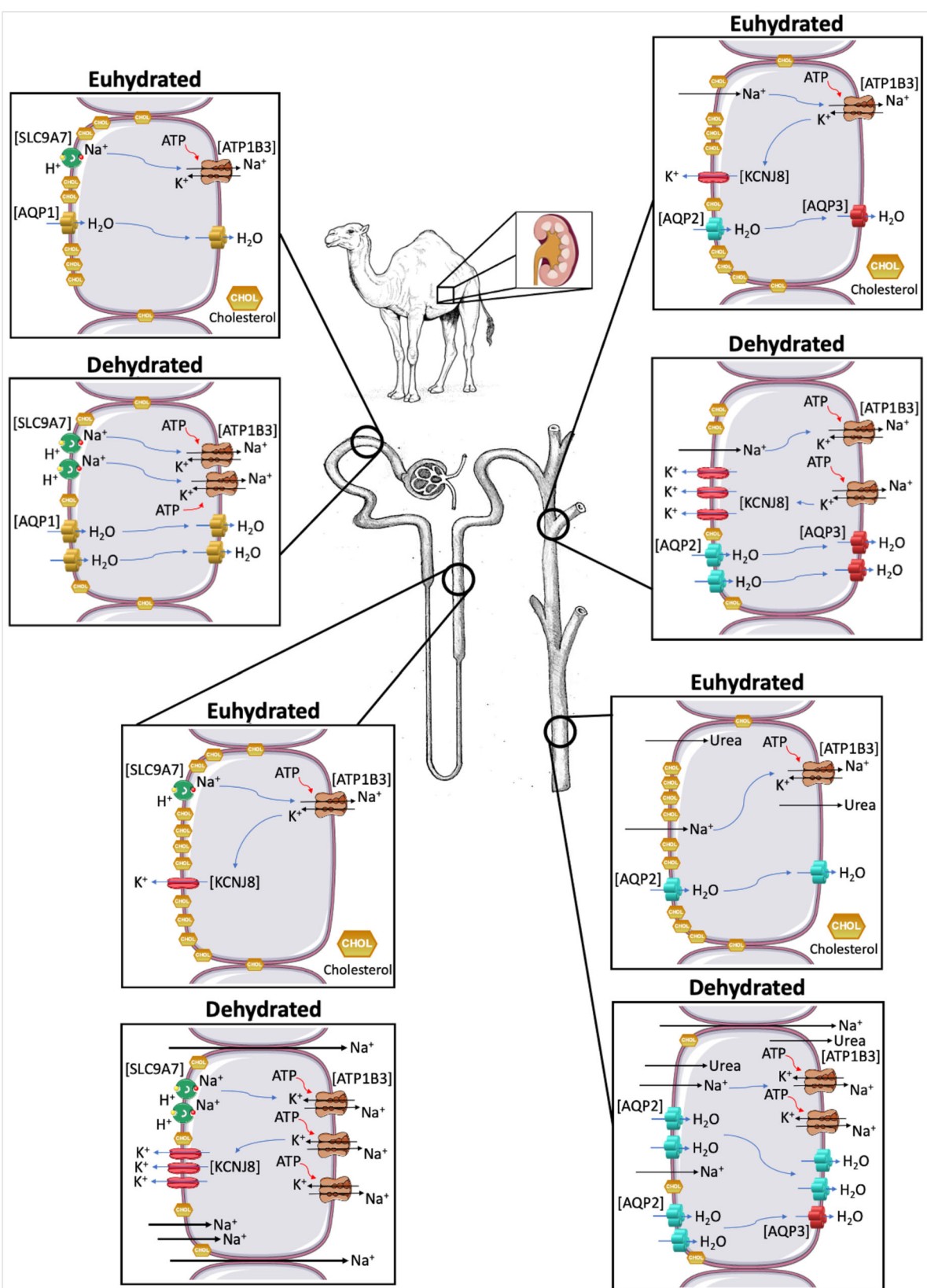

**Fig. 8 Ion and water transport in the kidney.** Schematic representation of the changes in transcellular movement of ions and water in the kidney between euhydrated and dehydrated dromedaries as a result of an increase in the abundance/activity of ion and water transporters facilitated by a partial depletion of cholesterol. Tubule lumen of the cell is on the left side of the cell diagrams whereas the interstitium is on the right side. All icons were freely available to download, modify and use in scientific publications from SMART Servier Medical Art under the terms of the Creative Commons Attribution 4.0 International License (https://smart.servier.com/) and from REACTOME under the Creative Commons Attribution 3.0 Unported License (https://reactome.org/icon-lib).

**Gene validation**. We validated key genes involved in the cholesterol biosynthesis pathway as well as ion and water transporters by RT-qPCR. Total RNA was extracted as mentioned above. cDNA was synthesised (GoScript™ Reverse Transcriptase, Promega) from 700 ng of total RNA following the manufacturer's protocol. RT-qPCRs were run using StepOne real-time qPCR system (Thermo Fisher Scientific) using PowerUp SYBR Green Master Mix (Thermo Fisher Scientific) in 10 µl reactions (final volume). We designed transcript-specific primer pairs using Primer-BLAST (NCBI)[131] (Supplementary Table ST5) and a dromedary reference genome (accession No.: PRJNA310822). We generated standard curves for each primer pair from serial dilutions (2-fold) and calculated $R^2$ and reaction efficiencies using the equation $E = 10-(1/\text{slope})-1$[132]. We finally determined transcription levels of these genes by calculating $\Delta\Delta C_t$ change corrected for reaction efficiency. We used PPIA as housekeeping gene after we shortlisted potential candidates[133].

**Cholesterol analysis**. Measurements of the levels of cholesterol in kidney tissue were performed at the Bristol Metabolomics Facility, University of Bristol. We used the same samples as for the transcriptomic and proteomic analyses and the RT-qPCR validation. A sample of 100–250 mg of frozen powdered tissue was provided to the Metabolomics Facility and the analyses were carried out using Cholesterol/Cholesterol Ester-Glo™ Assay kit (Promega) following manufacturer's protocol. Briefly, samples were weighed into Eppendorf tubes and freeze-dried overnight. Dry weight was recorded. Separately, a cholesterol standard at 80 µM was prepared by diluting 1 µl of cholesterol standard (20 mM) in 249 µl of lysis solution. Further standards at 20, 40 and 60 µM were prepared by further dilutions with lysis solution ($R^2 > 0.99$). Then, 100 µl of thawed and mixed lysis solution was added to each sample, and samples were incubated in a water bath at 37 °C for 30 min. Twenty microliters of each sample and standards were transferred to a 96-well white-walled assay plate together with 20 µl of Cholesterol Detection Reagent with esterase. Samples were incubated at room temperature for 1 h and the luminescence was recorded immediately after using a plate-reading luminometer. Total cholesterol is calculated by comparison of the luminescence of standards.

Serum cholesterol was measured using a fully automatic chemistry analyser (Roche/Hitachi Cobas c 311, USA). Blood samples were collected in plain vacutainers via jugular venipuncture, without apparent discomfort to the camels, between 8:00 and 9:00 am. The blood was allowed to clot for 30 min at room temperature and then, serum separated by centrifugation at $1790 \times g$ (Allegra X-15R Centrifuge, Beckman Coulter) for 10 min. Aliquots were transferred to clean vials and kept at −80 °C for later analyses. Serum was incubated with cholesterol esterase to yield free cholesterol. Then, cholesterol oxidase was added to catalyse the oxidation of cholesterol to cholest-4-en-3-one and $H_2O_2$. In the presence of peroxidase, $H_2O_2$ triggers the oxidative coupling of phenol and 4-aminophenazone to form a red quinone-imine dye. The intensity of the dye, which is directly proportional to the cholesterol concentration, was measured in a colorimeter at 700:505 nm.

Membrane cholesterol was quantified using a colorimetric-based Cholesterol/Cholesteryl Ester Quantification Assay kit (Abcam; ab65359). Briefly, the crude membrane fraction was enriched using a modified version of the plasma membrane isolation method described by Suski et al.[134]. Further fractionation of the crude plasma membrane fraction was not required since organelle membranes are well-known to be devoid of cholesterol[135,136]. Thus, we homogenised 1 g of ground kidney (cortex and medulla) tissue in 8 ml of isolation buffer 1 (IB-1 buffer) by vortexing and then proceeded with the protocol as described by the authors. Firstly, we centrifuged the homogenate at $800 \times g$ and 4 °C for 5 min (Sorvall Legend RT), this step was repeated with the supernatant. Then, we performed 2 centrifugations at $10,000 \times g$ and 4 °C for 10 min (Heraeus Biofuge Fresco) and discarded the pellets. Finally, the supernatant was spun 2 more times at $25,000 \times g$ and 4 °C for 30 min (Beckman Coulter Optima™ LE-80K Ultracentrifuge) to pellet the membranes. A 100 µl aliquot of the crude membrane fraction resuspended in 1 ml of starting buffer (SB buffer) was kept after the first $25,000 \times g$ centrifugation for the quantification of membrane protein. Crude plasma membranes where immediately resuspended in 200 µl of Chloroform:Isopropanol:NP-40 (7:11:0.1) for cholesterol extraction and spun at $15,000 \times g$ for 10 min at 4 °C. The liquid, organic phase was transferred to a clean tube. The samples were air-dried at 50 °C for 15 min to evaporate the chloroform and the organic solvent was removed using a mild $N_2$ stream. Samples were then dissolved in 155 µl of assay buffer. Reaction wells and the reaction mix were prepared as per the manufacturer's protocol using 50 µl of sample and 50 µl of reaction mix and measured in a colorimeter in duplicates, together with the standards, at OD570 nm. The amount of cholesterol was normalised against the amount of membrane protein which was quantified using the Bradford assay.

We further used Filipin III (Santa Cruz Biotechnology) staining for cholesterol visualization. Kidney, frozen sections of 20 µm were washed in PBS, stained with Filipin III for 30 min and then rewashed with PBS two more times. The slides were air-dried protected from light to avoid bleaching and then mounted using ProLong Gold™ antifade reagent (Invitrogen). The slides were examined in a widefield microscope (Leica DMI8 with CODEX, Leica). No signs of bleaching were detected. To facilitate sample comparison, intensity surface plots were rendered using a FIRE LUT and the surface plot tool in ImageJ.

**Statistics and reproducibility**. Benjamini–Hochberg was used to calculate significance levels in differential gene expression analysis. An independent $t$-test was used to compare peptide abundance ratios between conditions. These were corrected for multiple testing using a Benjamini–Hochberg adjustment. Differentially expressed genes (DEGs) are defined as those showing significant differential expression between two conditions, where significance threshold is reached when the adjusted $p$ value is below 0.05. The same applies to the differentially expressed proteins (DEPs). Statistical significance between RT-qPCR experimental groups was calculated using one-way ANOVA with Turkey's post hoc test for multiple pairwise comparisons. Same analyses were used to assess significant differences in the levels of cholesterol. Alternatively, we used Kruskal–Wallis test combined with Benjamini–Hochberg method for groups which followed non-normal distribution. All the outliers highlighted in the boxplots were included for the statistical analyses and the source data is available in Supplementary Data 9. Samples sizes ($n$) are described in the main text for every experiment/analysis. RT-qPCRs and cholesterol measurements for every individual sample were run in duplicates. All statistical tests were run using R software. $p_{adj} < 0.05$ was considered significant except for the analyses of the expression profiles where a threshold of $p_{adj} = 0.01$ was used.

**Reporting summary**. Further information on research design is available in the Nature Research Reporting Summary linked to this article.

## Data availability
The data underlying the transcriptomic analyses, including raw FASTQ files, bulk RNAseq counts, DESeq2 data and project metadata, have been deposited in NCBI's Gene Expression Omnibus[137] and are accessible through GEO Series accession number GSE173683. The mass spectrometry proteomics data have been deposited to the ProteomeXchange Consortium via the PRIDE[138] partner repository and can be accessed with the accession number PXD025644. Source data underlying figures are presented in Supplementary Data 8. All other data are available from the corresponding authors upon reasonable request.

## Code availability
All software and code used to analyse the data are described in the 'Methods' section, have been previously described in the literature and are common, well-established tools used in omics studies.

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

## Acknowledgements

This research was generously supported by grants from the Leverhulme Trust (RPG-2017-287) to B.T.G., F.A.-I., D.M. and M.P.G., and the United Arab Emirates University (UAEU)-Program for Advanced Research (UPAR-31M242) to A.A. Students were supported by grants from the Biotechnology and Biological Sciences Research Council-SWBio DTP programme (BBSRC BB/M009122/1) to B.T.G., the Medical Research Council (MRC 1662603) to A.P. and the British Heart Foundation (BHF FS/17/60/33474) to A.G.P.

## Author contributions

D.M. and M.P.G. conceived the project. D.M., M.P.G. and A.A. equally supervised the project. M.P.G. collected the samples. D.M., M.P.G. and B.T.G. designed the experiments. B.G. and F.A. analysed the transcriptomic and proteomic data and performed most bioinformatic analyses. F.A. performed the majority of the laboratory analyses and wrote the manuscript. A.P. was responsible for read alignment and data curation. P.L. run laboratory analyses. A.G.P. assisted with bioinformatic analyses. A.H.A., M.A.A. and N.H. performed laboratory and data analyses. P.B. performed the reference genome assembly and assisted with bioinformatic advice. All authors contributed to revise the manuscript.

## Competing interests

The authors declare no competing interests.
