## [Transparent Peer Review File · Communications Biology]

Reviewers' comments:

Reviewer #1 (Remarks to the Author):

In this manuscript, the authors study the one-humped Arabian Camel's kidney using complementary approaches to understand the molecular mechanism behind water conservation in non-model species. The manuscript is well-written, and the different sections of the manuscript are well-explained and easy to follow. The authors did a very nice job presenting the results clearly (given the vast amount of information they had to handle). I think the manuscript has at least three solid pillars that differentiate it from other studies carried out in non-model species concerning water conservation using "omics" analyses: (i) they analyzed the medulla and cortex of the kidney separately; (ii) they worked with a rehydrated phase and (iii) they carried out also proteomic analyses. After an initial analysis of the kidney gene expression, the authors focused their attention on cholesterol biosynthesis genes (this pathway was found to be overrepresented in RNA-seq and proteomic analysis in dehydrated animals) and validated their expression using RT-PCR. Then, they formulate the hypothesis about the partial depletion of cholesterol and the increase in the abundance of key ions and water transporters. In the discussion, they argue in favor of this hypothesis, deeply revisiting previous works. Overall, I think it is a very good manuscript. My only major concern is related to the limited number of transporters analyzed by the authors.

The authors focus the attention on four ions and one water transporter gene. The authors understand, and I agree, that these four genes are key to the nephron's countercurrent multiplication process. But what about other transporters (e.g., other water/ion/solute transporters + other ATPases)? Are they differentially expressed? Are they also upregulated in the severely dehydrated camel?. If more transporters are found to be upregulated, I think that the hypothesis would be strengthened (or at least the link between cholesterol and transporters).

Minor comments:

Why is the Dehydrated vs. Rehydrated condition not present in Figure 1f? From this Figure, I thought that this condition was not analyzed.

Line 153: I don't know where the 88% and 30 % come from.

A brief and general description of the proteomic results would help put the DEGs results in context (e.g., the total number of proteins found, the correlation with transcriptome data, Etc.).

Line 334: According to Supp. Table ST2, MVK was also found differentially expressed between Dehyd vs. Rehyd condition.

The hypothesis suddenly appears in line 395. I think the hypothesis should be mentioned earlier in the manuscript.

Parenthesis never close in lines: 514, 645.

Line 558: should read "Na⁺-K⁺-2Cl⁻".

Please double-check the correspondence between Ensembl IDs and "external gene name." In the main text, AQP2 is correctly associated with ENSCDRG00005005216, but in Table SD_19, AQP2 is wrongly associated with ENSCDRG00005001140.

Reviewer #2 (Remarks to the Author):

Gillard et al describe an interesting study in which the kidneys from camels are compared after different conditions. As expected with non-model species, there are limitations to sample collection, which leads to high between-sample variation and more noise in the analysis, but the obtained results are interesting. The patterns that emerge (such as decreased cholesterol biosynthesis during dehydration) can be used for hypothesis generation, which the authors do in their manuscript. Unfortunately, this is where it ends and most of the manuscript is an attempt to support the hypothesis by real-time PCR (which kind of works), cholesterol measurements (which does not show a difference) and a long speculative discussion.

Other comments:

One major problem I had with this manuscript was readability. Especially the Results section is challenging to read and the manuscript contains sentences that are difficult to follow. I would suggest trying to make it more reader friendly.

Several times it is mention that "we identified those genes with known roles in the countercurrent multiplication process in the kidneys", but it is unclear which those are. Are these KCNJ8, SLC9A7, and ATP1B3?

I might have missed this, but since the countercurrent multiplication process takes mostly place in the loop of Henle of the juxta medullary nephrons, would the authors not expect their RT-qPCR data from the medulla to track better than from the cortex? Their data shows the opposite and therefore weakens support for their hypothesis.

Having processes that take place in different cells shown in the same cell is confusing (Figure 7) and it would be better when in both A and B two cells (CD and TAL) would be drawn and showing the processes in the individual cells. Also euhydrated and dehydrated should be clearly labeled in the figure (instead of using A and B)

The authors mention being concerned about their RNA quality. Using an Agilent 2100 bioanalyzer would give a lot of insight into quality and degradation. RIN should be provided for each sample to ensure there is no bias in RNA quality towards a specific group.

I was surprised by the comment that "some animals being clearly older than others" while the Methods section says all animals were between 4 and 5 years old. Please explain.

Minor comments:

1. DEG needs to be spelled out in the abstract.
2. In figure 2 the same order of C, D, and R should be used in the graph and the heatmaps
3. Gene names always need to be in italics (standard nomenclature)
4. Instead of using A and B in figures 5 and 6, it would be easier if the graphs would be clearly labeled with 'cortex' and 'medula'
5. Genbank accession number is missing
6. Proteomic data does not seem to have been deposited in a public database.

Reviewer #3 (Remarks to the Author):

In the manuscript entitled "A comprehensive analysis of gene expression ----- role for cholesterol in water conservation during dehydration" Gillard et al made a heroic effort to comprehensively catalogue the transcriptome and proteome of the dromedary camel kidneys. The investigation presented in the manuscript are very valuable in the context of climate change and thus will help us to understand the mechanisms of water control in dehydration. The manuscript is well written with minor recommendations.

1) In the supplementary data sets SD1 onward, it will be very helpful to the readers if the authors add a column providing gene name. A simple VLOOKUP function in Excel data sheets will do the job in one stroke.

2) The authors used RIPA buffer to homogenize the samples for protein isolation followed by TMT tagging and mass spec. I am very puzzled by the negligible membrane proteins in the samples. RIPA contains tons of SDS and Triton X-100 which will very efficiently solubilize membrane proteins. Our experience with TMT method following a simple digestion in TAEB buffer containing 1% SDS was able to identify almost all of the membrane proteins. Probably such harsh centrifugation at 14000 g may not be required. Centrifugation at 5000 g for 15 min will be sufficient to remove debris and nuclei. It would be great if the authors repeat the proteomic analysis. I do understand that it is a costly protocol and maybe helpful for future studies.

3) Line 354, it should be "ACAT2" not "ACTA2".

4) A major goal of the manuscript is to elucidate the role of cholesterol in water conservation. The authors report no change in cholesterol yet they make the claim that a decrease in cholesterol favors increased membrane expression of Aqp2 and other membrane transporters involved in water conservation. If this hypothesis is to be addressed, the authors must measure cholesterol concentration in plasma membranes or at least in crude membranes. There are several very good methods for enriching plasma membranes from un-perfused rat and mouse kidneys which maybe suitable for camel kidneys.

COMMSBIOL Revisions – Rebuttal letter for Referees

Manuscript number: COMMSBIO-21-0102A

We would like to thank the referees for their robust and scholarly assessment of our manuscript. The suggested revisions have greatly improved the paper. All changes are highlighted in red in the revised manuscript.

Review #1

1. My only major concern is related to the limited number of transporters analyzed by the authors. (e.g., other water/ion/solute transporters + other ATPases)? Are they differentially expressed? Are they also upregulated in the severely dehydrated camel?

Whilst we have considered other transporters, we initially decided to validate and focus only on those known to be affected by the levels of cholesterol in order to present the data in the clearest way possible. We acknowledge that important information may have been left out as a consequence. Thus, we revisited the RNAseq data and identified all differentially expressed water/ion/solute transporters (aquaporins, sodium, potassium, chloride, ATPases, all members of the Slc gene family). Then, we investigated their functions and identified those previously described as being involved in the urinary concentrating process.

In addition to the genes already mentioned in the manuscript, we have now included and validated AQP1, AQP3 and ENaC. Figure 6 has been modified accordingly. Other previously described transporters were investigated as well but showed no significant change or were not detected in the RNAseq data as shown in Table 2. Figure 7 has also been modified to include AQP1 and AQP3 and these genes, together with their potential modulation by changes in the level of cholesterol, has been discussed.

2. Why is the Dehydrated vs. Rehydrated condition not present in Figure 1f? From this Figure, I thought that this condition was not analyzed.

As mentioned by the reviewers, we worked very hard to ensure the clearest way to present the data. The reviewer mentions, nevertheless, a very good point that we carefully considered and was the topic of long discussions whilst designing the figures. We concluded that including a 3rd comparison (Reh vs Deh) in a 3 way Venn plot was very confusing and did not help towards understanding the data. Rather we decided to present the 2 conditions under study (dehydration and rehydration) compared to control animals to focus on differentially expressed genes that change under these two conditions compared to euhydrated animals.

A Reh vs Deh comparison is, however, very important and that is why we came up with the idea of presenting them as heatmaps, clustering genes with equal expression profiles (Figure 2). Additionally, we have now included numbers of genes that were only differentially changed in rehydration compared to dehydration. In this way, one can easily spot how genes behave as if presented in a time series, thus allowing the inference of the Reh vs Deh comparison. In order to make the reader able to find out whether a specific gene has significantly changed between these 2 conditions, we have included 4 additional supplementary data files (cortex/medulla – RNA/protein) containing gene/protein catalogues of differentially expressed genes/proteins exclusively found in rehydration compared to dehydration. Thus, we believe that using Fig 2 and the supplementary datasets makes it easier to identify genes of interest and their specific expression profile.

3. Line 153: I don't know where the 88% and 30 % come from.

88% of the DEGs identified in the cortex after dehydration returned to control expression level. Likewise, 30% of the DEGs found in the medulla in dehydrated animals returned to control expression levels.

We have modified the text accordingly to make this clear. It now reads “Our data shows that 1139 genes were differentially expressed in the dromedary kidney during dehydration compared to control. Of all DEGs identified in dehydrated animals, 88% and 30%, returned to control expression levels in the cortex and the medulla, respectively. We identified a larger number of DEGs during dehydration than during rehydration in both tissues.”

4. A brief and general description of the proteomic results would help put the DEGs results in context (e.g., the total number of proteins found, the correlation with transcriptome data, Etc.).

We have added the suggested information in the results section about the RNA protein overlap (Line 279).

“In total, we identified 1282 genes to be differentially expressed in at least one tissue and one condition. Similarly, we catalogued 3002 proteins with significantly changed peptide abundance. Interestingly, further comparisons between RNAseq and proteomic data revealed that only 157 DEGs had their corresponding protein differentially expressed.”

5. Line 334: According to Supp. Table ST2, MVK was also found differentially expressed between Dehyd vs. Rehyd condition.

We thank the referee for drawing our attention to this error; the RNAseq data (ST2) shows that MVK is significantly downregulated in rehydration compared to control in the medulla, not in dehydration compared to control as was written in the manuscript. The has been corrected.

However, it is not differentially expressed in rehydration vs dehydration (Table ST2).

6. The hypothesis suddenly appears in line 395. I think the hypothesis should be mentioned earlier in the manuscript.

We have more clearly stated our hypothesis at the end of the introduction (Line 117), where we briefly summarise the work presented in the paper and now clearly state the hypothesis formulated based on that work. Additionally, we have included our hypothesis after the transcriptomic and proteomic sections to pinpoint that the cholesterol biosynthesis pathway is consistently enriched in our datasets (Line 347), which also helps with the narrative and makes clear why we validate the genes mentioned in the following sections.

7. Parenthesis never close in lines: 514, 645.

This error has been corrected. The text now reads “Under normal conditions, plasma cholesterol in one-humped Arabian camels has been reported to be approximately 400 µg/mg (calculated from Mohri et al 68) and it was...”

and

“...creating the osmotic gradient widely described as the “single effect” (for review; 37).”

8. Line 558: should read "Na⁺-K⁺-2Cl⁻".

Amended.

9. Please double-check the correspondence between Ensembl IDs and "external gene name." In the main text, AQP2 is correctly associated with ENSCDRG00005005216, but in Table SD_19, AQP2 is wrongly associated with ENSCDRG00005001140.

All datasets in the supplementary data have been double-checked while fulfilling comment 1 of reviewer #3. We ran biomaRt and g:Profiler to assign gene names to Ensembl gene IDs and that process has been repeated to assure no erroneous associations are present.

Review #2

1. One major problem I had with this manuscript was readability. Especially the Results section is challenging to read, and the manuscript contains sentences that are difficult to follow. I would suggest trying to make it more reader friendly.

We have done our best to make the manuscript “more reader friendly”. Specifically, we have gone through the most complex and long sentences in the Results section and tried to simplify the text.

2. Several times it is mention that “we identified those genes with known roles in the countercurrent multiplication process in the kidneys”, but it is unclear which those are. Are these KCNJ8, SLC9A7, and ATP1B3?

We thank the referee for her/his comment. We agree that the phrasing “we identified those genes with known roles in the countercurrent multiplication process in the kidneys” is mentioned numerous times in the text with limited explanation.

That specific sentence is mentioned in the abstract, in the results section and in the first paragraph of the discussion. We believed that adding information about these genes and clearly stating that these are our genes of interest in the Results section, where they are first mentioned (apart from the abstract), is the best option. We rephrased the sentence to clearly state this and add information about the function of those genes to stress their importance. Together with the extensive sections about them in the discussion, we think the narrative is easier to follow.

Thus, the text starting in line 446 reads “Based on our hypothesis of a role for cholesterol during dehydration, we identified those genes with known roles in the countercurrent multiplication process in the kidney which were potentially affected by changes in the level of cholesterol in the cell/cell membrane and showed differential expression in the RNAseq data. These genes are KCNJ8 (Potassium inwardly rectifier; ENSCDRG00005018658), SLC9A7

(Solute carrier; H⁺ / Na⁺ symporter; ENSCDRG00005020882), ATP1B3 (Na⁺ / K⁺ - ATPase; ENSCDRG00005009339), and the gene coding for the water channel Aquaporin 2 (AQP2; ENSCDRG00005005216). KCNJ8 pumps K⁺ ions from the cell into the lumen, SLC9A7 transports Na⁺ ions into the cell, ATP1B3 actively pumps Na⁺ into the interstitium and AQP2 allows water to flow through the cell into the interstitium to be reabsorbed.”

3. I might have missed this, but since the countercurrent multiplication process takes mostly place in the loop of Henle of the juxta medullary nephrons, would the authors not expect their RT-qPCR data from the medulla to track better than from the cortex? Their data shows the opposite and therefore weakens support for their hypothesis.

Additional analyses showing a decrease in membrane cholesterol in both cortex and medulla have further clarified the role of cholesterol during dehydration. Thus, we believe that the RT-qPCR validations give a good idea of what is happening at the gene expression level. Perhaps not as many genes coding for enzymes are significantly downregulated in the cortex as in the medulla but we argue that, overall, the pathway is blunted. Especially important is the significant downregulation of the major rate-limiting enzyme in cholesterol biosynthesis, HMGCR, in both tissues (references provided in the manuscript). Together with the decrease in membrane cholesterol we believe that we provide sufficient data to confidently say that the pathway is suppressed in both, cortex and medulla.

Thus, despite high variability in expression of some of the genes assessed by RT-qPCR, taken together, our RNAseq data, proteomics, RT-qPCR validation and membrane cholesterol quantification and visualization suggest that the suppression of genes involved in cholesterol biosynthesis and the subsequent reduction in membrane cholesterol are a global response in the kidney of the one-humped Arabian camel to dehydration, affecting both cortical and medullary segments.

Nevertheless, we acknowledge that further research, perhaps repeating similar studies after microdissecting the kidney nephron, will be necessary to precisely localise where changes in cholesterol are most prominent within the camel nephron.

The text has been modified to emphasise these points in the paragraph starting in line 384, line 415, line 424, the paragraph starting in line 428, line 516, line 520 and in the paragraph starting in line 720.

4. Having processes that take place in different cells shown in the same cell is confusing (Figure 7) and it would be better when in both A and B two cells (CD and TAL) would be drawn and showing the processes in the individual cells. Also euhydrated and dehydrated should be clearly labelled in the figure (instead of using A and B)

Figure 7 is now Figure 8 and has been modified according to the Referee's comment and the additional information that we provide regarding membrane cholesterol quantification. We have separated the processes and they now display in 8 different cells corresponding to the Proximal Convolute Tubule, TAL and the cortical and medullary CD (euhydrated and dehydrated states) as suggested by the reviewer. The labels "euhydrated" and "dehydrated" have also been included. The caption and the reference to the figure in the main text have been modified accordingly.

5. The authors mention being concerned about their RNA quality. Using an Agilent 2100 bioanalyzer would give a lot of insight into quality and degradation. RIN should be provided for each sample to ensure there is no bias in RNA quality towards a specific group.

Rather than being concerned about RNA quality, we wanted to stress that given the technical difficulties of collecting camel kidneys, RNA degradation might be higher than when working with a model species in a laboratory controlled environment. However, RIN showed that the quality of our RNA is well within acceptable limits (8.1 ± 0.5 , 7.7 ± 0.2 and 7.6 ± 0.6 for control, dehydrated and rehydrated cortex samples, and 8.5 ± 0.5 , 8.8 ± 0.2 and 8.7 ± 0.4 for control, dehydrated and rehydrated medulla samples, respectively). Perhaps we did not express it in the most appropriate way.

The referee suggests, and we agree, that RINs should be provided for each sample. Thus, we have included means \pm SD in the main text for each group ("Material and methods" section)

and a table (shown below) with values for each sample in the supplementary material. We further ran pairwise comparisons between groups to assure no bias towards specific groups (no statistical differences were found; $p > 0.05$).

Cortex			Medulla		
Sample Name	RIN	28/18S	Sample Name	RIN	28/18S
CC1	7.8	1.6	MC1	8.8	1.6
CC2	7.5	1.7	MC2	9.1	1.6
CC3	8.2	1.8	MC3	8.0	1.4
CC4	8.7	2	MC4	8.4	1.6
CC5	8.4	2	MC5	8.1	2.1
Mean	8.1		Mean	8.5	
SD	0.5		SD	0.5	
CD1	7.6	1.9	MD1	9.0	1.7
CD2	8	1.8	MD2	8.6	1.6
CD3	7.6	1.8	MD3	8.7	1.7
CD4	7.8	1.7	MD4	8.9	1.6
CD5	7.4	1.5	MD5	9.0	1.5
Mean	7.7		Mean	8.8	
SD	0.2		SD	0.2	
CR1	8.5	1.5	MR1	8.6	1.8
CR2	7.9	1.5	MR2	8.9	1.8
CR3	7.1	1.5	MR3	8.0	1.7
CR4	7.1	1.7	MR4	8.9	1.7
CR5	7.5	2	MR5	9.0	1.5
Mean	7.6		Mean	8.7	
SD	0.6		SD	0.4	

We have made this clearer in the “Limitations of the study” section as well.

- I was surprised by the comment that “some animals being clearly older than others” while the Methods section says all animals were between 4 and 5 years old. Please explain.

Differences were clear in terms of weight but not age, it was an unintended mistake to refer to both in the sentence mentioned by the reviewer. It has been corrected and now reads “We detected differences in weight, with some animals being clearly heavier than others, which could alter the degree of response to dehydration. Further, despite all camels being in the age range of 4-5 years, the small variation could influence the ability to respond to dehydration.”

7. Minor comments:

1. DEG needs to be spelled out in the abstract.

It now reads “Thus, we further identified differentially expressed genes with known roles in...”

2. In figure 2 the same order of C, D, and R should be used in the graph and the heatmaps.

We thank the referee for her/his comment and acknowledge that having the conditions displayed in the same order would be ideal. However, heatmaps arrange the samples according to similarity of their gene expression profile and the dendrogram splits branches following that criteria as well. Thus, as far as we know, it is not within our reach to arrange the heatmaps in a specific manner. Rearranging the samples would cause the dendrogram branches to crossover one another.

3. Gene names always need to be in italics (standard nomenclature).

All gene names in the main text are now italicised. We are aware uppercase gene names are used for human genes; however, the convention is that camel genes shall be written in uppercase (all) and italics (as you suggested) (Jirimutu et al 2012). Protein names are as usual in uppercases (not italicised).

1. Instead of using A and B in figures 5 and 6, it would be easier if the graphs would be clearly labeled with ‘cortex’ and ‘medula’.

As suggested, a and b labels in fig 5 and 6 have been replaced by “cortex” and “medulla” for clarity. Additionally, we realised font sizes were perhaps too small so have increased those in titles and axis to make it easier to read. The figure caption and the main text have been amended accordingly.

5. Genbank accession number is missing.

The data underlying the transcriptomic analyses, including raw FASTQ files, bulk RNAseq counts, DESeq2 data and project metadata, have been deposited in NCBI's Gene Expression Omnibus and are accessible through GEO Series accession number GSE173683 at <https://www.ncbi.nlm.nih.gov/geo/query/acc.cgi?acc=GSE173683>.

Data will remain private until publication of the manuscript, but the Referees may access it via the link above using the following token to get access: wxcbiwkelrwxzmd

6. Proteomic data does not seem to have been deposited in a public database.

The mass spectrometry proteomics data have been deposited to the ProteomeXchange Consortium via the PRIDE partner repository with the dataset identifier PXD025644. The dataset will remain private until the paper is accepted for publication, but reviewers can access the data using the following credentials.

Username: reviewer_pxd025644@ebi.ac.uk

Password: 4juY12fw

Review #3

1. In the supplementary data sets SD1 onward, it will be very helpful to the readers if the authors add a column providing gene name. A simple VLOOKUP function in Excel data sheets will do the job in one stroke.

Using biomaRt and gProfiler (g:Convert gene ID conversion tool) we have added a column with gene names and a second one with gene description. Additionally, we have added the same information to the protein catalogues where one can now find Ensembl transcript IDs, Ensembl gene IDs, gene names and gene description. Please, note that the gene name column will display an Ensembl gene ID when gene names are missing (e.g., novel genes). We considered using g:Ortholog to provide Human or mouse gene orthologue names, but this approach introduces duplicates since one Camel Ensembl gene ID may match 2 or more Human/mouse orthologues due to the relatively incomplete annotation of the camel genome. Thus, we decided to leave those genes without a name and solely provide Ensembl gene IDs.

2. The authors used RIPA buffer to homogenize the samples for protein isolation followed by TMT tagging and mass spec. I am very puzzled by the negligible membrane proteins in the samples. RIPA contains tons of SDS and Triton X-100 which will very efficiently solubilize membrane proteins. Our experience with TMT method following a simple digestion in TAEB buffer containing 1% SDS was able to identify almost all of the membrane proteins. Probably such harsh centrifugation at 14000 g may not be required. Centrifugation at 5000 g for 15 min will be sufficient to remove debris and nuclei. It would be great if the authors repeat the proteomic analysis. I do understand that it is a costly protocol and maybe helpful for future studies.

We appreciate the comment from Referee #3 and will most certainly take his/her advice into account for future experiments. As he/she correctly mentions, it is, unfortunately, not possible to rerun additional proteomic analyses due to the costs and the fact that the funding for this project is ending shortly, however we agree it would greatly benefit the study.

Centrifugation at ~12000g is widely recommended in the literature for both proteomics and Western blot using a variety of buffers (Zeng and Kentsis 10.21203/rs.3.pex-1036/v1, <https://bit.ly/3gRzUOW>), with some protocols even using substantially higher speeds. Thus, we were, to the best of our knowledge, using an appropriate protocol. Unfortunately, as the Referee points out, there are better options and that is likely the reason why we missed a fraction of proteins. Further, despite RIPA is routinely used to solubilise proteins, a fraction is RIPA-insoluble so that may be also affecting the extraction. All these issues will be considered in future studies.

These issues have been stressed in the "Limitation of the study" section.

3. Line 354, it should be "ACAT2" not "ACTA2".

The incorrect name has been replaced by the correct one.

4. If this hypothesis is to be addressed, the authors must measure cholesterol concentration in plasma membranes or at least in crude membranes. There are several very good methods for enriching plasma membranes from un-perfused rat and mouse kidneys which maybe suitable for camel kidneys.

We first measured serum cholesterol to confirm that the results we had obtained from measuring kidney cholesterol were biased by the presence of blood in the sample. Once we confirmed this, we moved on to measure membrane cholesterol as suggested by the Referee.

We have used differential centrifugation to isolate plasma membranes and measured cholesterol in this preparation using a cholesterol assay kit. Thus, we invested considerable effort to optimise the protocol for frozen camel kidney samples. In the process, we tested different amounts of starting material and measured protein at several steps during the membrane isolation process to assure reliability. Once we were confident with the methodology and obtained consistent, solid results we ran the whole set of samples. We normalised the cholesterol measures to the concentration plasma membrane protein in the

same isolation. Final analyses were run using 3 biological replicates per group. Increasing this number was not feasible due to the limited amount of sample remaining from this study.

The following text has been added to the manuscript in the Results section, line 415, and the Material and Methods section, line 931.

Serum cholesterol was measured using a fully automatic chemistry analyser (Roche/Hitachi Cobas c 311- USA). Blood samples were collected in plain vacutainers via jugular venipuncture, without apparent discomfort to the camels, between 8:00 and 9:00 am. The blood was allowed to clot for 30 minutes at room temperature and then, serum separated by centrifugation at 4000 rpm for 10 minutes. Aliquots were transferred to clean vials and kept at -80°C for later analyses. Serum was incubated with cholesterol esterase to yield free cholesterol. Then, cholesterol oxidase was added to catalyse the oxidation of cholesterol to cholest-4-en-3-one and H₂O₂. In the presence of peroxidase, H₂O₂ triggers the oxidative coupling of phenol and 4-aminophenazone to form a red quinone-imine dye. The intensity of the dye, which is directly proportional to the cholesterol concentration, was measured in a colorimeter at 700:505 nm.

Membrane cholesterol was quantified using a colorimetric-based Cholesterol/Cholesteryl Ester Quantification Assay kit (Abcam; ab65359). Briefly, the crude membrane fraction was enriched using a modified version of the plasma membrane isolation method described by Suski et al. Further fractionation of the crude plasma membrane fraction was not required since organelle membranes are well-known to be devoid of cholesterol. Thus, we homogenised 1 gram of ground kidney (cortex and medulla) tissue (prepared as explained in Material and Methods section, line 847) in 8ml of isolation buffer 1 (IB-1 buffer) by vortexing and then proceeded with the protocol as described by the authors. Firstly, we centrifuged the homogenate at 800xg and 4°C for 5 minutes (Sorvall Legend RT), this step was repeated with the supernatant. Then, we performed 2 centrifugations at 10000xg and 4°C for 10 minutes (Heraeus Biofuge fresco) and discarded the pellets. Finally, the supernatant was spun 2 more times at 25000xg and 4°C for 30 minutes (Beckman Coulter Optima™ LE-80K Ultracentrifuge) to pellet the membranes. A 100µl aliquot of the crude membrane fraction resuspended in 1ml of starting buffer (SB buffer) was kept after the first 25000xg

centrifugation for the quantification of membrane protein concentration. The final crude plasma membrane pellets were immediately resuspended in 200 μ l of Chloroform: Isopropanol: NP-40 (7:11:0.1) for cholesterol extraction and centrifuged at 15000xg for 10 minutes at 4°C. The liquid, organic phase was transferred to a clean tube. Samples were placed at 50°C for 15 minutes to evaporate chloroform and samples were evaporated to dryness under nitrogen gas. Samples were reconstituted in 155 μ l of assay buffer. The assay was performed as per the manufacturer's protocol using 50 μ l of sample and 50 μ l of reaction mix and measured in a colorimeter in duplicate, together with the standards, at OD570 nm. The amount of cholesterol was normalised against the amount of membrane protein which was quantified by Bradford assay.

We further used Filipin III (Santa Cruz Biotechnology) staining for cholesterol visualization. Kidney, frozen sections of 20 μ m were washed in PBS, stained with Filipin III for 30 minutes and then rewashed with PBS two more times. Sections were air dried in darkness to prevent bleaching and coverslipped using ProLong Gold™ antifade reagent (Invitrogen). Sections were examined on a widefield microscope (Leica DMI8 with CODEX). To perform comparisons of signal intensity, intensity surface plots were rendered using a FIRE LUT and the surface plot tool in ImageJ.

Membrane cholesterol was reduced in dehydration compared to control in both cortex and medulla, although variability prevented the differences from reaching the significance threshold. In the cortex, the membrane cholesterol concentration was 418 \pm 127 μ g/mg of protein (n=3) in control animals and decreased to 270 \pm 129 μ g/mg of protein (n=3) in dehydrated animals. Similarly, in the medulla, membrane cholesterol concentration was 353 \pm 108 μ g/mg of protein (n=3) and 256 \pm 40 μ g/mg of protein (n=3) in control and dehydration, respectively (Fig 6c). We further imaged kidney sections under the widefield microscope using the cholesterol-specific marker Filipin III and generated intensity surface plots to compare the intensity of the fluorescence signal. Signal was reduced in dehydrated samples compared to controls in tubular cells of both tissues (Fig 7d-e). Taken together, these results are supportive of changes to cholesterol synthesis and possibly a reduction in membrane cholesterol along the nephron of the kidney.

REVIEWERS' COMMENTS:

Reviewer #2 (Remarks to the Author):

The authors have done a wonderful job addressing my comments and suggestions and feel this is ready for publication.

Reviewer #3 (Remarks to the Author):

The authors have addresses all the concerns I raised. No more comments